# Stretching Beyond the Obvious: A Gradient-Free Framework to Unveil the Hidden Landscape of Visual Invariance

**Lorenzo Tausani**[1]**, Paolo Muratore**[1]**, Morgan B. Talbot**[2,3,4]**, Giacomo Amerio**[1]**,
Gabriel Kreiman**[2,3]**, Davide Zoccolan**[1]
[1]Neuroscience Area, International School for Advanced Studies (SISSA), Trieste (Italy)
[2]Boston Children's Hospital, Harvard Medical School, Boston (USA)
[3]Center for Brains, Minds, and Machines, MIT, Cambridge (USA)
[4]Harvard-MIT Program in Health Sciences and Technology, MIT, Cambridge (USA)

## Abstract

Uncovering which feature combinations are encoded by visual units is critical to understanding how images are transformed into representations that support recognition. While existing feature visualization approaches typically infer a unit's most exciting images, this is insufficient to reveal the manifold of transformations under which responses remain invariant, which is critical to generalization in vision. Here we introduce Stretch-and-Squeeze (SnS), an unbiased, model-agnostic, and gradient-free framework to systematically characterize a unit's maximally invariant stimuli, and its vulnerability to adversarial perturbations, in both biological and artificial visual systems. SnS frames these transformations as bi-objective optimization problems. To probe invariance, SnS seeks image perturbations that maximally alter (stretch) the representation of a reference stimulus in a given processing stage while preserving unit activation downstream (squeeze). To probe adversarial sensitivity, stretching and squeezing are reversed to maximally perturb unit activation while minimizing changes to the upstream representation. Applied to CNNs, SnS revealed invariant transformations that were farther from a reference image in pixel-space than those produced by affine transformations, while more strongly preserving the target unit's response. The discovered invariant images differed depending on the stage of the image representation used for optimization: pixel-level changes primarily affected luminance and contrast, while stretching mid- and late-layer representations mainly altered texture and pose. By measuring how well the hierarchical invariant images obtained for $L_2$-robust (i.e., adversarially trained) networks were classified by humans and other observer networks, we discovered a substantial drop in their interpretability when the representation was stretched in deep layers, while the opposite trend was found for standard (i.e., not robustified) models. This indicates that $L_2$ adversarial training fails to increase the interpretability of high-level invariances, despite good perceptual alignment between humans and robustified models at the pixel level. This demonstrates how SnS can be used as a powerful new tool to measure the alignment between artificial and biological vision.

## 1 Introduction

Both visual neuroscience and deep learning seek to understand image processing systems composed of millions of interacting functional units, whose activity patterns are shaped by their experience with natural image statistics (Matteucci et al., 2024; Barrett et al., 2019). This common goal raises a fundamental question in both fields: which combination of image features do visual neurons become tuned for? Traditionally, this question has been addressed by developing feature visualization approaches that discover the "preferred" stimuli that maximally activate a given unit - often referred to as the unit's most exciting images (MEIs) (Olah et al., 2017; Nguyen et al., 2019; Fel et al., 2023; Xiao & Kreiman, 2019; Walker et al., 2019). MEIs, however, only reveal a few instances within the

vast set of images that strongly activate a unit (Nguyen et al., 2016; Cadena et al., 2018), offering poor insight into the manifold of transformations under which the unit's activity remains invariant.

To overcome this limitation, we developed Stretch-and-Squeeze (SnS), an unbiased, model-agnostic method to probe visual invariance in both artificial and biological neurons. Building upon an existing framework for finding MEIs (Xiao & Kreiman, 2020), SnS optimization exploits evolutionary algorithms, but with a different objective: to find invariant or adversarial images. SnS integrates the search for invariant images and adversarial examples within a unified optimization objective, generating image perturbations starting from a reference image (e.g., a MEI of the unit). To probe a unit's invariance, SnS explicitly looks for images that are maximally distinct from this reference stimulus in the representation of a chosen visual processing stage, while preserving the response of a target unit downstream. This allows SnS to sample the invariance manifolds of the selected unit when the representation of an effective stimulus (either a MEI or a natural image) is stretched at different processing stages. As a result, our approach reveals the actual image variation axes the unit is able to tolerate, providing a richer, more veridical description of its invariance landscape, as compared to traditional tests based on predefined (e.g., affine) transformations (Goodfellow et al., 2009; Kheradpisheh et al., 2016; Engstrom et al., 2019c). In our study, we carried out comprehensive tests of the effectiveness of SnS to achieve the goals listed above, using a popular CNN (ResNet50) as a benchmark. To study invariances of the whole model with respect to specific image classes, we applied SnS to the network's readout units, as they are selective for the object categories the model has learned to classify (e.g., a 'cup' neuron). We found dramatic differences among the invariance landscapes of these units when stretching the representations at different depths of the processing hierarchy. Stretching early, middle, and late representations yielded invariant images that differed from the reference (and among themselves) in terms of luminance/contrast, texture, and pose, respectively. We also discovered important differences in these hierarchical invariances between the standard and an adversarially trained version of the network. Like previous studies (Feather et al., 2023) we found that invariant images from the $L_2$-robust network were more recognizable by human subjects and other *observer* networks. However, we identify a striking point of divergence: while robust CNN invariances become less interpretable when stretching at deeper layers, those of standard networks become more intrepretable, causing the robust network's advantage to erode in later layers. Importantly, being gradient-free and model-agnostic, SnS can uncover the invariance fields of units in "black-box" image processing systems, where access to hidden units is either absent or very limited. Neurophysiologists face this scenario when they want to infer the tuning properties of biological visual neurons. To test the applicability of SnS in neurophysiology experiments, we verified that the method works even if the experimenter can record the activity of just a small fraction of the units in the processing stage where the stretching is applied.

## 2 RELATED WORKS

**Probing CNN representations: feature visualization, invariance, and adversarial examples.** The question of functional interpretability and feature visualization in deep learning has predominantly been approached by employing gradient-based optimization for image synthesis, taking advantage of the complete analytical description and differentiability of the network (Olah et al., 2017; Nguyen et al., 2019; Fel et al., 2023). While most studies have focused on finding the MEIs of CNNs' units, recent efforts have started to also explore their invariance landscapes. Most of them have focused on providing a local measure of model invariance around a given input, probing the shape of the representation manifold in the vicinity of a chosen image (Berardino et al., 2017; Hénaff & Simoncelli, 2015). Another prominent approach relies on discovering *model metamers* - i.e., synthesized stimuli that match the internal representation of a reference (natural) image in a specific layer of a network (Mahendran & Vedaldi, 2015). Using metamers, Feather et al. (2023) were able to demonstrate that standard CNNs display highly idiosyncratic invariances at the top layers of processing, with metamers being unintelligible to human observers or other neural networks. In contrast, CNNs trained to be robust to adversarial images (i.e., imperceptibly modified inputs capable of altering CNN object classification (Szegedy et al., 2013)) yielded metamers that were substantially more interpretable by human observers. This shows how invariance and adversarial vulnerability are conceptually related (Jacobsen et al., 2020). For instance, adversarially trained ("robust") networks can craft subtle image perturbations that not only fool the network but can also impair (Gaziv et al., 2023) or enhance (Talbot et al., 2025) human object recognition, thereby reinforcing the perceptual parallels between robust network representations and invariance in human vision.

**Applications to visual neuroscience.** Advances in CNN interpretability have also impacted neuroscience, which has long suffered from a lack of effective methods to investigate the functional tuning of visual neurons in higher cortical areas. By leveraging the strong functional analogy between CNNs trained for image classification and the primate object recognition pathway (known as the *ventral stream* (DiCarlo et al., 2012)), it is possible to create *digital twins* of the ventral stream using CNNs (Yamins et al., 2014; Yamins & DiCarlo, 2016; Schrimpf et al., 2018). This allows employing gradient-based optimization to synthesize images that modulate the activity of biological neurons (Bashivan et al., 2019) or investigate properties like adversarial robustness in visual neurons (Guo et al., 2022). The digital twin paradigm has also been extended beyond primate vision, informing models of the auditory cortex (Kell et al., 2018) and visual processing in rodents (Nayebi et al., 2023; Walker et al., 2019; Tong et al., 2023). Importantly, related gradient-based techniques (Cadena et al., 2018) have also been used to map invariance in individual neurons, for instance, by looking for images that maximally activate a mouse primary visual neuron while being maximally distinct in pixel space (Ding et al., 2023). Other recent approaches (Baroni et al., 2023; Bashiri et al., 2025) used Compositional Pattern-Producing Networks (CPPNs) to map low-dimensional (1 and 2d) invariance manifolds of digital twin neurons of macaque primary visual cortex, successfully recovering visual invariances established by previous neurophysiological and computational work (Schwartz et al., 2006; Sharpee, 2013).

Obviously, these approaches have an intrinsic limitation: they are only as good as the fidelity of the digital twin at capturing the selectivity of visual neurons. To overcome this constraint, gradientless feature visualization methods like XDREAM have successfully synthesized effective MEIs for units in both the primate ventral stream (Ponce et al., 2019) and artificial neural networks (Xiao & Kreiman, 2019) by relying on evolutionary algorithms. While recent work has also begun to explore the feature landscape around the MEIs obtained with XDREAM (Wang & Ponce, 2022b), current gradientless approaches have not been systematically applied to characterize the invariance of visual tuning in artificial and biological architectures. Hence the novelty of SnS, which is, to the best of our knowledge, the first gradientless approach to systematically infer the invariance manifolds of visual units.

## 3 METHODS

### 3.1 THE STRETCH-AND-SQUEEZE ALGORITHM (SNS)

SnS consists of three key components: a generative model $\psi$, a test (or *subject*) network $\phi$ and a gradient-free optimizer (Fig. 1a). The generative model is a pretrained deep neural network (Dosovitskiy & Brox, 2016) that maps $n$-dimensional vectors $\boldsymbol{\xi}^t \in \mathbb{R}^n$ (referred to as *codes*) to RGB images: $\boldsymbol{x} = \psi(\boldsymbol{\xi}) \in \mathcal{X} \subseteq \mathbb{R}^{C \times H \times W}$. In all the experiments, $n = 4096$. Crucially, the generative model $\psi$ was trained on naturalistic stimuli and embodies a powerful prior over the distribution of possible images. We use the Covariance Matrix Adaptation Evolutionary Strategy (CMA-ES) optimizer (Hansen et al., 2003) to adjust the codes and iteratively improve on our objective. At each iteration $t$ the optimizer yields a new set of codes $\boldsymbol{\xi}^{t+1} \in \mathbb{R}^n$, which in turn is used to generate a new batch of images. A core tenet of the SnS algorithm is the relational construction of its fitness function. We start by introducing two layer indices $\kappa$ and $\ell$ for our test network $\phi$, where for convention we include the input stage as $\kappa = 0$, and the measuring function $\Gamma(\boldsymbol{x}, \phi^\ell) \in \mathbb{R}^d$, which returns the activations $\mathbf{a}^\ell$ of all the $d$ units in layer $\ell$, when the network $\phi$ is presented with stimulus $\boldsymbol{x}$. We then identify a reference stimulus $\boldsymbol{x}_{\mathrm{ref}}$ from which we construct the pair of reference states $\left(\mathbf{a}_{\mathrm{ref}}^\kappa, \mathbf{a}_{\mathrm{ref}}^\ell\right)$ as $\mathbf{a}_{\mathrm{ref}}^{\kappa,\ell} = \Gamma\left(\boldsymbol{x}_{\mathrm{ref}}, \phi^{\kappa,\ell}\right)$. Lastly, we introduce two optimization objectives as either the minimization $\mathcal{L}_{\mathrm{squeeze}}$ or maximization (i.e. minimization of the negative) $\mathcal{L}_{\mathrm{stretch}}$ of the euclidean distance of a given state $\mathbf{a}^\kappa = \Gamma(\boldsymbol{x}, \phi^\kappa)$ from the corresponding reference state:

$$\mathcal{L}_{\mathrm{stretch}}^\kappa\left(\mathbf{a}^\kappa, \mathbf{a}_{\mathrm{ref}}^\kappa\right) = -\parallel \mathbf{a}^\kappa - \mathbf{a}_{\mathrm{ref}}^\kappa \parallel_2, \quad \mathcal{L}_{\mathrm{squeeze}}^\kappa\left(\mathbf{a}^\kappa, \mathbf{a}_{\mathrm{ref}}^\kappa\right) = +\parallel \mathbf{a}^\kappa - \mathbf{a}_{\mathrm{ref}}^\kappa \parallel_2. \quad (1)$$

Then, for a given choice of layer indices $\kappa, \ell$ and reference states $\mathbf{a}_{\mathrm{ref}}^{\kappa,\ell}$, indicating with $\mathcal{X} \subseteq \mathbb{R}^{C \times H \times W}$ the set of input images, we define $f : \mathcal{X} \to \mathbb{R}^2$ as:

$$f : \boldsymbol{x} \mapsto \begin{pmatrix} \mathcal{L}^{\kappa}_{\text{stretch}} \left( \Gamma\left(\boldsymbol{x}, \phi^{\kappa}\right), \mathbf{a}^{\kappa}_{\text{ref}} \right) \\ \mathcal{L}^{\ell}_{\text{squeeze}} \left( \Gamma\left(\boldsymbol{x}, \phi^{\ell}\right), \mathbf{a}^{\ell}_{\text{ref}} \right) \end{pmatrix}, \tag{2}$$

and introduce the bi-objective minimization problem: $\min_{\boldsymbol{\xi}} \left( f_1(\boldsymbol{x}), f_2(\boldsymbol{x}) \right)$, where $\boldsymbol{x} = \psi\left(\boldsymbol{\xi}\right)$.

We define $\Xi_{\text{SnS}}$ as the formal solution to such bi-objective optimization problem in the Pareto sense, i.e. $\Xi_{\text{SnS}}$ is the collection of Pareto-optimal solutions $\boldsymbol{x}^* \in \mathcal{X}$. In particular, denoting a solution $\boldsymbol{x}_1$ $f$-dominant with respect to $\boldsymbol{x}_2$ by $\boldsymbol{x}_1 \succ_f \boldsymbol{x}_2$, where we write $\boldsymbol{x}_1 \succ_f \boldsymbol{x}_2 \iff f_i(\boldsymbol{x}_1) \leq f_i(\boldsymbol{x}_2) \forall i$ and $\exists i$ for which the inequality holds strictly, we can write the $\Xi_{\text{SnS}}$ collection of solutions as:

$$\Xi_{\text{SnS}} \equiv \left\{ \boldsymbol{x}^* \in \mathcal{X} : \left\{ x \in \mathcal{X} : \boldsymbol{x} \succ_f \boldsymbol{x}^* \right\} = \varnothing \right\}. \tag{3}$$

In practice, for each round of optimization, we sorted the fitness scores by organizing them in Pareto fronts (Deb, 2011) (see Supplementary Material, Section A.3 for further details). In order to make our formulation more transparent to different components, we introduce the following shorthand notation for $\Xi_{\text{SnS}}$ and refer to equation 3 for its exact definition:

$$\Xi_{\text{SnS}} \equiv \arg\min_{\boldsymbol{x} \in \mathcal{X}} \left[ \mathcal{L}^{\kappa}_{\text{stretch}} \left( \Gamma(\boldsymbol{x}, \phi^{\kappa}), \mathbf{a}^{\kappa}_{\text{ref}} \right), \mathcal{L}^{\ell}_{\text{squeeze}} \left( \Gamma(\boldsymbol{x}, \phi^{\ell}), \mathbf{a}^{\ell}_{\text{ref}} \right) \right]. \tag{4}$$

While this formulation is general and in principle supports arbitrary choices for the layer indices $\kappa$ and $\ell$, as well for the corresponding reference states $\mathbf{a}^{\kappa,\ell}_{\text{ref}}$, in our experiments we restricted the scope of the optimization problem to investigate specific properties of the test network $\phi$ (e.g. robustness, invariance) by making the following design choices. First, we singled out a target unit $u^{\ell}_{\text{targ}}$ in layer $\ell$ (see below for details) and considered $\boldsymbol{x}^{\star} \in \mathcal{X}$ the maximally exciting stimulus (i.e., the MEI) for our target unit, computed via 500 iterations of the XDREAM algorithm (Ponce et al., 2019). Our reference state for layer $\ell$ was then $a^{\ell}_{\text{ref}} = \Gamma(\boldsymbol{x}^{\star}, \phi^{\ell}_u) \in \mathbb{R}$, i.e. the scalar activation of the target readout unit for its MEI. We then explored the set of solutions $\Xi_{\text{SnS}}$ as we varied $\kappa < \ell$. In particular, we can express both the search for adversarial attacks or invariant stimulus for our target unit $u^{\ell}_{\text{targ}}$ as the following two optimization problems:

$$\Xi_{\text{inv}} \equiv \arg\min_{\boldsymbol{x} \in \mathcal{X}} \left[ \mathcal{L}^{\kappa}_{\text{stretch}} \left( \Gamma\left(\boldsymbol{x}, \phi^{\kappa}\right), \Gamma(\boldsymbol{x}^{\star}, \phi^{\kappa}) \right), + \left| a^{\ell}_u - a^{\ell}_{\text{ref}} \right| \right] \tag{5}$$

$$\Xi_{\text{adv}} \equiv \arg\min_{\boldsymbol{x} \in \mathcal{X}} \left[ \mathcal{L}^{\kappa}_{\text{squeeze}} \left( \Gamma\left(\boldsymbol{x}, \phi^{\kappa}\right), \Gamma(\boldsymbol{x}^{\star}, \phi^{\kappa}) \right), - \left| a^{\ell}_u - a^{\ell}_{\text{ref}} \right| \right], \tag{6}$$

where we have used $\Gamma\left(\boldsymbol{x}, \phi^{\ell}_u\right) = a^{\ell}_u$ and wrote $\mathcal{L}^{\ell} = \pm \left| a^{\ell}_u - a^{\ell}_{\text{ref}} \right|$.

This dual formalization reflects the fact that both invariance and robustness relate changes of high order representations to changes at the input level (Fig. 1b). However, we remark that a key flexibility of SnS is that its general formalization for the *stretch* and *squeeze* objectives allows computation not only in the input pixel space, but also in any intermediate representation stages $\kappa$ and $\ell$ of the network. This allows probing invariance (or adversarial attacks) at different levels of feature abstraction.

For the characterization of invariance, we set this representation space at three distinct hierarchical levels within ResNet50: (*i*) $\kappa = 0$, i.e., the input pixel space (denoted `low_level`), (*ii*) a mid-level convolutional layer (1$^{\text{st}}$ convolutional stage in layer 3, `mid_level`), and (*iii*) a deep convolutional layer (7$^{\text{th}}$ convolutional stage of layer 4, `high_level`). Our primary target units $u^{\ell}_{\text{targ}}$ were chosen in the final readout layer (fully connected), as these units are selective for specific object categories, allowing us to study the invariance of the whole network to transformations of the image classes those neurons are tuned to. To demonstrate SnS's generalizability beyond category-selective units, we also performed optimizations targeting units within `mid_level` and `high_level`, which are potentially more analogous to biological neurons in the visual system. In these cases, the input-side distance was computed in pixel space (i.e. $k = 0$). For adversarial attack generation, we restricted our analysis to the configuration targeting readout layer units and defining input distance in pixel space. SnS optimizations were conducted on the units of two specific instances of ResNet50, our subject network: the standard ImageNet-pretrained model available in PyTorch (Paszke et al., 2019), and a robust counterpart generated via adversarial training with an $L_2$ perturbation norm constraint

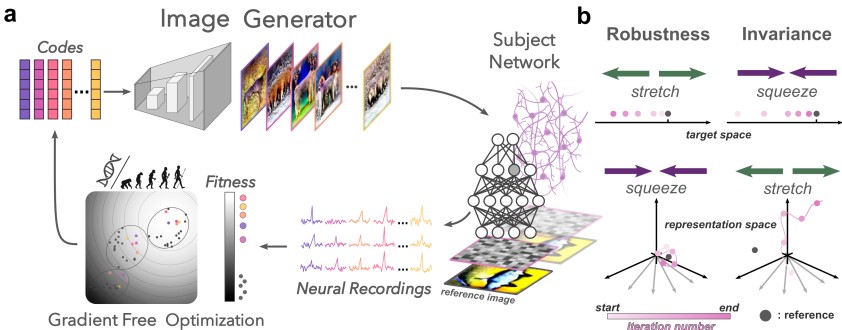

Figure 1: **The Stretch-and-Squeeze (SnS) algorithm.** (**a**) Overview of the SnS algorithm: candidate stimuli are synthetized from latent codes via the image generator, and the activation response of target units is recorded and used as fitness score by the optimizer, that adjusts a new set of codes. (**b**) Dual fitness objectives in SnS. To probe *invariance* (right), SnS maximizes stimulus distance from a reference in the representation space (stretch) and minimizes the variation in the activation of a target unit (squeeze). Conversely, to synthesize *adversarial examples* (left), SnS minimizes changes in the representation space, while maximizing the variation in the activation of the target unit.

of $\epsilon = 3$ (Engstrom et al., 2019a). SnS invariances were studied also in ResNet18 and VGG16_bn (see Section 3.2) and Vision Transformers (ViT, see Supplementary Material, Section I).

Initialization strategies were adapted based on the optimization goal: for adversarial attacks, the search was initialized from the MEI $x^\star$, for invariance experiments, initialization began from random normally distributed vectors. For additional details regarding the computational experiments we refer to Supplementary Material, Section A.

### 3.2 INTERPRETABILITY OF THE INVARIANT IMAGES BY HUMANS AND OBSERVER NETWORKS

To assess whether the invariant images produced by SnS retain sufficient information for correct classification by other visual systems, we tested the ability of humans or other *observer* neural networks to classify them, using a 12-alternative forced choice task (AFC). We used several distinct ImageNet-pretrained architectures as observer networks (ResNet18, VGG16_bn, Wide-ResNet50-2, DenseNet161, ResNeXt50_32x4d, Shufflenet_v2_x1_0), both standard obtained from PyTorch (Paszke et al., 2019), and robust (all $L_2$, $\epsilon = 3$), obtained from Salman et al. (2020). To assess human recognizability of invariant images, we recruited 25 human participants using the online Prolific platform. Details on the procedures to run these interpretability experiments are reported in Supplementary Material, Section B.

## 4 RESULTS

### 4.1 SNS GENERATES EFFECTIVE ADVERSARIAL AND INVARIANT IMAGES

We first validated the SnS framework by generating invariant and adversarial images for a sample of 77 readout units of a $L_2$-robust ResNet50 using their MEIs $x^\star$ as reference images, and applying the stretching (to achieve invariance) or the squeezing (to achieve adversarial images) to the pixel representation (see Section 3.1). For each unit, 10 independent invariant and adversarial images were synthesized, each with a different initialization seed, and distance metrics with respect to the reference MEI were aggregated as the mean over the synthesized images. These metrics were then further aggregated across units as mean $\pm$ SD and reported in Fig. 2. As shown in the figure, SnS successfully generated effective adversarial examples (top left). These stimuli substantially suppressed the activation of the readout units relative to their MEIs (mean reduction of $111\% \pm 7\%$), being displaced from the MEIs by a mean $L_2$ distance of $72 \pm 12$ pixels. This relatively large pixel budget reflects both the network robustness and our stringent unit-silencing objective - a stricter criterion than mere misclassification. Consistent with Tsipras et al. (2018) and Gaziv et al. (2023), perturbations were semantically relevant, not noise-like (Fig. 2).

The invariant images generated by SnS (bottom right of Fig. 2) were also very effective, yielding only a minor drop of the units' activation relative to the MEIs (mean reduction of $34\% \pm 11\%$). At the same time, they substantially departed from the MEIs, with a mean $L_2$ pixel distance ($271 \pm 32$ pixels) that considerably exceeded the median distance between ImageNet images, as reported by Gaziv et al. (2023). More impressively, SnS discovered image transformations that were more extreme (in terms of pixel distance) than those achieved by applying to the MEI standard data augmentations (colored dots/lines), while altering the activation of the readout units substantially less than the most extreme augmentations (darker dots). This indicates that SnS can discover the actual axes of image variation a unit learned to tolerate, exploring invariance manifolds more precisely than traditional tests with predetermined (e.g., affine) transformations. The same result was obtained when we applied SnS to discover invariant and adversarial images for a subset of readout units, but taking highly effective natural images as references instead of the MEIs (see Fig. S5a). See also Fig. S5b for additional examples of invariant and adversarial images obtained using as references either MEIs or natural images. The ability to obtain invariant and adversarial images is enabled intrinsically by the dual loss formulation of the SnS optimization. Indeed, when images are synthesized using only one of the two loss components at a time, results diverge drastically from those of SnS (see Fig. S6 in Section D of the Supplementary Materials).

Finally, we note that SnS successfully generated invariant images also for units in intermediate convolutional layers (Supplementary Material, Section E). This demonstrates its applicability to the analysis of latent representations that are more analogous to those of biological neurons.

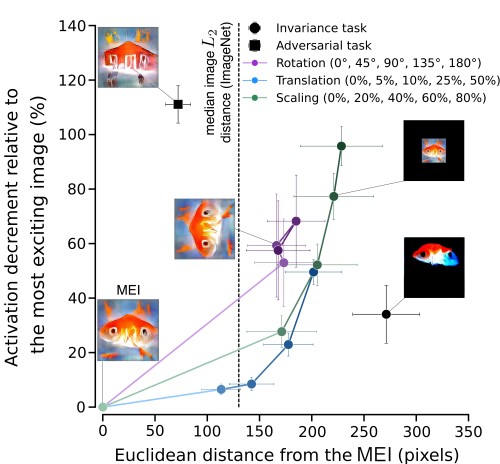

Figure 2: **SnS generates effective adversarial and invariant examples.** The average activation reduction ($\pm$ SD) of $n = 77$ readout units of a $L_2$-robust ResNet50 (relative to their MEIs) is plotted against the $L_2$ pixel distance from the MEIs, when the latter were transformed with SnS to yield either adversarial or invariant images, or were subjected to affine transformations (rotation, translation and scaling). The pixel distance refers to $224 \times 224$ RGB images with values between $0$ and $1$.

## 4.2 SnS REVEALS LAYER-SPECIFIC INVARIANT MANIFOLDS

Next, we compared the kinds of invariant images that SnS discovered by stretching the MEI representations of the 77 readout units at different processing stages along a $L_2$-*robust* ResNet50 - i.e. in the pixel space (as done in Fig. 2), as well as in an intermediate (`Layer3_conv1`) and a deep (`Layer4_conv7`) convolutional layer. Qualitatively, this yielded different sorts of invariances (Fig. 3a). Stretching in the pixel-space produced mainly luminance and contrast changes; stretching mid-level representations mainly affected texture and color; stretching high-level representations tended to produce abstract variations like viewpoint changes or multiple object instances. The same qualitative differences were observed when SnS was used to produce invariant images for 77 readout units of *standard* ResNet50 (Fig. 3b), although, in this case, the images appeared to be less interpretable than those obtained for the robust counterpart. Also, the images lacked the high-frequency patterns that are typically found in the MEIs of standard networks units (Engstrom et al., 2019b), suggesting that the SnS generator strongly regularizes the image search towards natural statistics.

The qualitative descriptions above are just approximate semantic labels to concisely describe image transformations that are, in fact, too rich and complex to be readily captured by a textual description (e.g., stretching in pixel space also produced size changes). To quantify these observations more objectively, we applied PCA to the invariant images yielded by SnS. We then used the first $k$ principal components as feature vectors to train a Support Vector Classifier (SVC) to classify the invariant images according to the layer where the SnS stretching was applied (for details see Supplementary Material, Section A.4). As shown in Fig. 3c, a few components were enough to yield above

Figure 3: **SnS discovers layer-specific invariances**. (**a**) Example MEI and associated invariant images obtained for a readout neuron ("cup") in $L_2$-robust ResNet50 and computed for three different choices of the stretching representational stages (rows). Multiple results are shown for different random initialization seeds (columns) (**b**) Same as (**a**) but for a standard ResNet50. (**c**) Accuracy of a SVC in discriminating the three classes of invariant images produced by stretching pixel-, mid-, and high-level representations as a function of the number of principal components fed to the classifier.

chance performance (i.e., $> 33.3\%$ correct), and a few tens of components enabled near-perfect discrimination for the standard network (red) and above $80\%$ correct for the robust network (green).

To characterize how the invariant images generated by stretching at a given processing stage were represented across the network, we measured their $L_2$ distance from their reference MEIs in the representation provided by each layer of the network (see Supplementary Material, Section A.5.2 for detail). In addition to demonstrating the effectiveness of SnS to explore invariances far away from their reference MEI and heterogeneous across multiple runs (see Section A.5.2 for detail), the representational distance between the invariant images and their MEIs was preserved in the layers that were adjacent to the one where the stretching was applied, revealing a clear hierarchical pattern (Fig. S3 a,b). Stretching in the pixel-space yielded invariant images that remained dissimilar from the MEIs in the first layers of the network; stretching in mid and deep convolutional layers yielded images that were more different from the MEIs in the central and final portions of the network.

Yet another way to characterize the invariance manifolds is to estimate their intrinsic dimension (ID). We did so for an example readout unit of $L_2$-robust ResNet50, using both PCA and 2NN-ID, a state-of-the-art nonlinear estimator (Facco et al., 2017) (see Section F in the Supplementary Material). 2NN-ID yielded ID estimates one order of magnitude lower than PCA, indicating that the geometry of the three manifolds was highly nonlinear (Fig. S8). Both methods, however, revealed a consistent ID ranking across stretching layers: lowest for the pixel space, highest for mid layers, and lower again for deep layers - a pattern mirroring previously reported trends in the ID of image representations across deep CNNs and visual cortex (Ansuini et al., 2019; Muratore et al., 2022a)

**Hierarchical invariances are robust to representation subsampling.** To assess SnS's applicability to neuroscience, where only a small neuronal subpopulation is typically recorded from a visual area, we generated the invariant images using heavily subsampled hidden layer representations. Both qualitative inspection and representational distance analysis revealed that the resulting invariances closely resembled those obtained using the full layer representation, supporting SnS's potential for neuroscience experiments (see Supplementary Material, Section G for further details).

### 4.3 INVARIANT IMAGES FOR ROBUST AND STANDARD NETWORKS ARE PERCEPTUALLY DIFFERENT

We next evaluated how recognizable the invariant images generated by SnS were by humans and other CNNs. In our tests, invariant images for both kinds of networks (i.e., standard and $L_2$-robust) were generated by stretching the low- (pixel), mid- and high-level representations of the same set of natural images sampled from 12 different Imagenet categories (Fig. 4a). Human subjects performed a 12-AFC classification task on these invariant images, their reference images, and MEI controls (Fig. 4a, top; for details see Section 3.2 and Supplementary Section B). Invariant images from the robust network (all generation stages) were significantly more recognizable by humans than their standard network counterparts (Fig. 4b, compare the green to the red boxes; p < 0.001, Supplementary

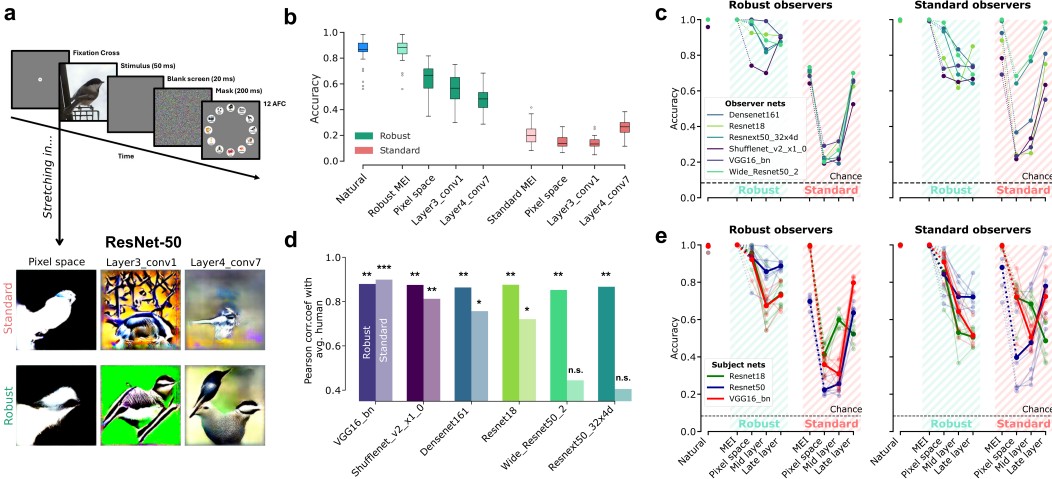

Figure 4: **Interpretability of the invariant images by humans and other networks.** (**a**) Illustration of the human classification task with the invariant images generated by SnS for standard and $L_2$-robust networks. (**b**) Classification accuracies of humans (averaged across subjects and categories) displayed for each experimental condition. (**c**) Classification accuracies across multiple $L_2$-robust (left) and standard (right) networks. (**d**) Correlation coefficient between average human performances and the performances of each observer model. For each architecture, the darker color indicates the robust model and the lighter color indicates the standard version. (**e**) Analysis from (**c**) extended to multiple architectures: ResNet50, ResNet18, and VGG16_bn. Solid lines indicate the average between different observer networks (translucent lines).

Material Table S2). Invariant images from the $L_2$-robust network were more recognizable when stretching was applied to earlier-layer representations, with pixel-space stretching yielding the highest discrimination accuracy. Interestingly, an opposite trend was observed for the standard network: invariant images from a late layer were more recognizable than those from a middle layer or from pixel space (p < 0.001 for each of the above comparisons: see Supplementary Material Table S2). These findings provide a more granular understanding of the perceptual misalignment between humans and CNNs, extending beyond what classic feature visualization tools can reveal with a direct comparison of MEIs. In fact, human participants classified the MEIs obtained for the robust network (Fig. 4b, cyan box) with an accuracy comparable to that achieved on the natural images encoded by the corresponding readout units (Fig. 4b, blue box). By comparison, human classification of the MEIs obtained for the standard network was relatively poor (Fig. 4b, pink box). Therefore, considering only MEIs' interpretability, the $L_2$ robustification process would seem very effective in increasing the alignment of image classification between ResNet50 and human observers. SnS reveals instead that this alignment gradually deteriorates when considering invariant images obtained by stretching the representation in progressively deeper layers, with the interpretability gap between the robust and standard networks progressively narrowing. Interestingly, this also shows how SnS can uncover trends that are complementary to those found by previous approaches aimed at assessing invariance, such as computation of metamers (Feather et al., 2023). Similarly, SnS provides a complementary tool to test model alignment with human perception with respect to previous approaches which study alignment through model disagreement (Wang & Simoncelli, 2008; Golan et al., 2020), as well as to other approaches that have examined human interpretability of parametric image transformations at the pixel or patch level (Ullman et al., 2016; Malik et al., 2023).

The same classification task was administered to various *observer* CNNs (both standard and $L_2$-robust). The alignment with the accuracy trends found in humans was relatively high for all robust observer networks (compare Fig. 4b to 4c,left), while their standard counterparts displayed a more heterogeneous degree of consistency with human choices (compare Fig. 4b to 4c,right). As a result, correlations with human perception were very high and statistically significant for all robust networks, but only for a fraction of the standard ones (Fig. 4d). Similarly to humans, all $L_2$-robust observer networks better classified the invariant images obtained for $L_2$-robust ResNet50 than for its standard

counterpart, with this difference narrowing for higher-order invariances. This demonstrates that, despite the $L_2$ robustification process, the invariance landscape of ResNet50 readout units remained idiosyncratic and not fully generalizable to other human and artificial observers. This conclusion held true also when standard networks were used as observers, although, for some models, the interpretability gap between invariant images obtained for robust and standard networks was narrower (Fig. 4c, right). Similar trends were found when SnS was used to uncover the hierarchical invariances of two different subject networks (ResNet18 and VGG16_bn, again $L_2$-robust and standard; Fig. 4e).

We also applied SnS to produce hierarchical invariant images for the readout units of $L_\infty$-robust ResNet50. In this case as well, we obtained layer-specific invariant manifolds that were easily separable in pixel space (Fig. S10a and b). However, unlike $L_2$-robustification, $L_\infty$-robustification yielded invariant images whose interpretability by $L_\infty$-robust observer networks remained consistently high or even increased with stretching depth (Fig. S10c). This indicates that $L_\infty$ adversarial training yields invariance manifolds that are, overall, more interpretable than those produced by $L_2$ robustification.

To demonstrate the flexibility of the SnS framework, we performed the same interpretability experiment on invariant images evolved for readout units of a vision transformer (ViT). Similar to CNNs, the nature and interpretability of ViT invariances depended on the hierarchical level at which the representation was stretched. Invariances generated from pixel space were substantially less interpretable than those from mid- and high-level representations, which were remarkably similar and more interpretable. This observation aligns with the view that ViTs learn less strictly hierarchical and more globally-integrated features than CNNs (Dosovitskiy et al. (2020); see Fig. S11 in Section I of the Supplementary Material). Taken together, the results of Fig. 4 and Fig. S11 indicate that the level of transferability of the invariances obtained for the readout units of a generator network depends on three key factors: 1) the architecture of the generator (i.e., CNNs vs. vision transformers); 2) its training procedure (robustified vs. standard, in the case of CNNs); and 3) the hierarchical stage at which the representation is stretched to produce the invariant images.

## 5 DISCUSSION

We introduced Stretch-and-Squeeze, a novel, model-agnostic framework that reconceptualizes how we probe invariance of visual representations. Instead of testing pre-defined transformations (e.g., affine) or imposing strict representational identity (as with metamers (Feather et al., 2023)), SnS discovers the actual manifold of transformations a unit tolerates by maximizing stimulus dissimilarity in a chosen representational space while preserving the target unit's activation. This exploration of functionally equivalent yet representationally distinct inputs allows SnS to map a broader, potentially more "ecologically" relevant stretch of a unit's response manifold and its boundaries. Compared to other approaches aimed at uncovering invariant manifolds (Cadena et al., 2018; Baroni et al., 2023), SnS is distinguished by its gradient-free and model-agnostic formulation, a critical advantage for applications in neuroscience, especially in cases where reliable digital twin models are unavailable. Moreover, differently from these previous approaches, which have been applied to explore only the low-dimensional invariance fields of visual neurons in early processing stages (i.e, primary visual cortex), SnS operates in a highly expressive search space (i.e., the latent code space with dimension $d = 4096$), which imposes only a soft constraint on the dimensionality of the invariant manifolds. This allows sufficient expressivity to effectively map the complex invariant fields of high-order units, such as CNNs' readout neurons, thus characterizing the invariances of the entire model with respect to the object categories it has learned to classify.

The hierarchical application of SnS shows that CNN units in readout layers are not necessarily highly robust to the standard transformations (scaling, translation, etc) often applied to probe invariance (DiCarlo et al., 2012; Goodfellow et al., 2009; Kheradpisheh et al., 2016; Engstrom et al., 2019c) (Fig. 2). Rather, they strongly tolerate image changes along radically different axes of abstraction (Fig. 3): from low-level properties (e.g., luminance) to mid-level features (e.g., texture) and, finally, to high-level semantic variations (e.g., object pose). This illustrates how the invariance is built hierarchically, and how this process is intertwined with the process of building feature selectivity in the first place: while sensitivity to increasingly complex combinations of visual features emerges across CNN layers, from the standpoint of a readout unit, it is the *insensitivity* to those same, increasingly complex visual properties that is progressively gained. In other words, while a visual

system strives to gain selectivity for increasingly complex visual features, it must also become invariant to those same features' combinations.

Our comparison of the invariant images generated for standard and robustly-trained ResNet50 extends prior work on the topic (Feather et al., 2023) with an innovative characterization. The superior semantic coherence and cross-system recognizability (by humans and other CNNs) of the invariances obtained for robust networks strongly corroborates that adversarial training sculpts representations towards human-aligned perceptual features (Engstrom et al., 2019b; Gaziv et al., 2023; Feather et al., 2023). However, unlike computation of MEIs or metamers (Feather et al., 2023), SnS reveals additional trends that are consistent across all human and CNN observers: there are opposite trends for the interpretability of the invariant images generated for the $L_2$-robust and the standard networks as a function of stretching depth (Fig. 4c, e). Specifically, we show that stretching low-level representations generates human-interpretable invariances in robustly trained networks, whereas it produces poorly interpretable invariances in standard networks. Conversely, stretching high-level representations enhances interpretability in standard networks but decreases it in $L_2$-robust networks. These trends are different from those obtained by testing the interpretability of metamers (see Table S3). This is not surprising, given that metamers are invariant images that minimize the distance from a target natural image in a given layer, rather than maximizing it, as SnS does. Thus, while SnS pushes the search for invariant images to the boundaries of the invariance manifold, metamers explore a much narrower neighborhood of the target image (see Table S4). Hence the complementary nature of the two approaches.

Finally, the model-agnostic and gradientless nature of SnS positions it as a powerful new tool for neuroscience, particularly for systems where high-fidelity "digital twins" may be hard to develop. Our demonstration of SnS's efficacy with simulated sparse neural recordings directly addresses a critical experimental constraint, paving the way for in vivo applications. This could lead to the discovery of new, hierarchical invariances in biological visual systems, extending our understanding of visual object representations in both primates (DiCarlo et al., 2012; Ponce et al., 2019) and other species (Muratore et al., 2025; 2022b; Tafazoli et al., 2017; Tong et al., 2023; Soto & Wasserman, 2016; Wood & Wood, 2015; Schluessel et al., 2012).

**Limitations and broader impacts.** While SnS has proven effective in uncovering novel, meaningful invariances, it presents opportunities for refinement and broader application. One avenue is to use SnS for further, more extensive geometric characterizations of the invariance manifolds, expanding the work presented in Supplementary Material, Section F. Another interesting direction is to apply SnS to characterize invariance in other modalities, such as audition, using a generator capable of producing audio waveforms or spectrograms from a low-dimensional latent code - e.g., SpecGAN, WaveGAN (Donahue et al., 2018). Returning to the visual domain, our work has demonstrated SnS' generative capabilities using eight different neural models (Fig. 4 and S11), belonging to three different architectural families, with parameter counts spanning roughly an order of magnitude (from 11.7M for ResNet-18 to 86.6M for ViT). This provides a strong test of our framework's generalizability, although future work could test SnS with even larger models (e.g., DINO (Caron et al., 2021), ViT22-B (Dehghani et al., 2023)). Finally, by pinpointing where the perceptual misalignment between robustified networks and humans is strongest (i.e., in higher-order invariances), SnS could guide the development of models that achieve more human-aligned invariance. For instance, one could design a visual diet where "good" (i.e., human-like) and "bad" (i.e., nonhuman-like) invariant images are explicitly used in the training of the network.

In terms of application to neuroscience studies, while SnS' robustness to significant representational subsampling provides a strong foundation for its neuroscientific applicability, the next step is direct validation in biological systems. Future research should focus on adapting SnS for in vivo experiments, addressing issues such as neural signal variability, the limited duration of neuronal recordings, and the need to optimize stimulus presentation in closed loop experiments, similarly to what was already achieved with XDREAM (Ponce et al., 2019). One challenge will be to leverage the wealth of neuronal data afforded by high-density silicon probes (Siegle et al., 2021), that can simultaneously record from hundreds to thousands of units. One strategy would be to apply SnS in parallel to multiple recorded neurons with non-overlapping receptive fields (RFs), thus independently evolving the distinct portions of the image processed by each unit. Conversely, for neurons with overlapping RFs, one could first screen them using natural images (Ponce et al., 2019) to search for subpopulations of units with similar selectivity. SnS could then be applied to the subpopulation to find the collective invariance field of the neuronal ensemble.

## 6 REPRODUCIBILITY STATEMENT

To enhance the reproducibility of our research, we provide the following resources. The SnS algorithm is described in detail within the main paper's Methods Section 3.1, with its corresponding pseudocode available in the Supplementary Material (see Algorithm 1). Furthermore, the complete source code used to conduct the experiments is available at https://github.com/zoccolan-lab/SnS, while experimental data were open sourced at Tausani et al. (2025). For clarity on our experimental setup, a general description of the experimental parameters can be found in the methods section. For a more granular understanding, further information on methodological details, hyperparameter settings and the computational resources used is reported in Supplementary Material, Section A. The Supplementary Material also includes a comprehensive section on the statistical analysis employed in our experiments (see Section B.1).

## 7 ETHICS STATEMENT

The human behavioral study was conducted under a pre-existing, approved Institutional Review Board (IRB) protocol. The task, which involved participants viewing images on a computer screen and making responses by clicking buttons, presented no more than minimal risk. Further details, including a complete reproduction of test instructions and information on participant compensation, are provided in Supplementary Material Section B.

## 8 ACKNOWLEDGEMENTS

We would like to thank Sebastiano Quintavalle for his help in implementing the code, and Eugenio Piasini for his suggestions on the Pareto front optimization. This work was funded by the European Union – NextGenerationEU – NRRM4C2-I.1.1, in the framework of the PRIN Project no. 2022WX3FM5, CUP:G53D23003220006 (DZ), by NIGMS Award T32GM144273 (MT), by NIH grant R01EY026025 (GK), and by the NSF Center for Brains, Minds, and Machines NSF CCF-1231216 (GK). The content of this paper is solely the responsibility of the authors and does not necessarily represent the official views of any of the above organizations.

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

# Supplementary Material

## A  ADDITIONAL DETAILS REGARDING COMPUTATIONAL EXPERIMENTS

### A.1  PSEUDOCODE FOR THE SNS ALGORITHM

Here we present the pseudo-code (Algorithm 1) for the `SnS` algorithm. We used library implementations for the `CMA-ES` gradient-free optimizer (Hansen et al., 2019) and the Pareto front computation (Fortin et al., 2012). Importantly, we introduce the following `should_stop` criteria for our experiments.

Optimizations were terminated by either reaching a maximum iteration limit (500) or satisfying an early stopping rule derived from the target unit's response statistics to a large dataset of natural images (i.e., the full ImageNet train set, $n \approx 1.2$ million). In particular, given the vastness of the dataset, we assumed that the activation elicited by the most effective image in the natural dataset ($a^\ell_{\mathrm{max\_nat}}$) represented a conservative threshold for an image to be considered *invariant*. On the other hand, the activation from the least effective image ($a^\ell_{\mathrm{min\_nat}}$) represents a non-response level, functioning as a threshold for an image to be considered *adversarial*. Therefore, when searching for adversarial images, termination was reached when a large fraction (i.e., $\geq 90\%$) of the current population of images elicited activations at or below $a^\ell_{\mathrm{min\_nat}}$, thus leading to a non-responsive status. Conversely, when looking for invariant images after initialization from random noise, it was terminated when a large fraction (i.e., $\geq 90\%$) of the population of images yielded activations equal or above $a^\ell_{\mathrm{max\_nat}}$, thus reaching the desired high-activation regime.

In terms of computational complexity, both generator-based image synthesis from latent codes and the extraction of activations from the target network require only a single feedforward pass per iteration, whose computational cost, while depending on network size, is negligible in practice for all the architectures examined in our study, which span roughly an order of magnitude of parameter counts (from 11.7M for ResNet-18 to 86.6M for ViT). Consequently, the dominant computational bottleneck is the CMA-ES optimizer, whose running time scales quadratically with the dimensionality $d$ of the search space. In practice, however, this does not pose a limitation: CMA-ES has been shown to be a highly effective method for gradient-free optimization of neural activity in vivo (Wang & Ponce, 2022a), and it remains computationally efficient under our hardware configuration as well (see Section A.6).

---

**Algorithm 1** SnS gradient-free optimization algorithm

---

**Require:** $\phi$          ▷ Target network
**Require:** $\psi$          ▷ Image generator
**Require:** $T \geq 0$          ▷ Maximum number of iterations
**Require:** $\kappa, \ell \in \{0, 1, \ldots, \texttt{depth}(\phi)\}$
**Require:** $\boldsymbol{x}_{\mathrm{ref}} \in \mathbb{R}^{C \times H \times W}$          ▷ Common choice is $\boldsymbol{x}_{\mathrm{ref}} = \boldsymbol{x}_\star$
     $t \leftarrow 0$
     $\mathbf{a}^{\kappa,\ell}_{\mathrm{ref}} \leftarrow \Gamma\left(\boldsymbol{x}_{\mathrm{ref}}, \phi^{\kappa,\ell}\right)$
     $\boldsymbol{\xi}_0 \leftarrow \mathcal{N}\left(0, \sigma_{\mathrm{init}}\right)$
     **while** $t \leq T$ and !early_stop **do**
         $\boldsymbol{x}_t = \psi\left(\boldsymbol{\xi}_t\right)$          ▷ Compute new candidate stimuli
         $\mathbf{a}^{\kappa,\ell} = \Gamma\left(\boldsymbol{x}_t, \phi^{\kappa,\ell}\right)$          ▷ Measure activations in target layers $\kappa, \ell$
         $\mathcal{L}^\kappa_{\mathrm{stretch}} \leftarrow - \parallel \mathbf{a}^\kappa - \mathbf{a}^\kappa_{\mathrm{ref}} \parallel$
         $\mathcal{L}^\ell_{\mathrm{squeeze}} \leftarrow + \parallel \mathbf{a}^\ell - \mathbf{a}^\ell_{\mathrm{ref}} \parallel$
         $\mathcal{P} \leftarrow \texttt{compute\_pareto\_front}\left(\boldsymbol{\xi}_t, \mathcal{L}^\kappa_{\mathrm{stretch}}, \mathcal{L}^\ell_{\mathrm{squeeze}}\right)$
         $\boldsymbol{\xi}_{t+1} \leftarrow \texttt{evolve}\left(\mathcal{P}, \boldsymbol{\xi}_t\right)$          ▷ Evolution strategy CMA-ES optimization
         $t \leftarrow t + 1$
         early_stop $\leftarrow \texttt{should\_stop}\left(\mathbf{a}^\kappa, \mathbf{a}^\ell\right)$
     **end while**

---

## A.2 Experimental Hyperparameters

The generative model $\psi$ is instantiated as a pretrained deep neural network variant, specifically the `fc7` configuration from Dosovitskiy & Brox (2016). The choice of this model instead of alternative models, such as as large-scale GANs and diffusion models, was motivated by the fact that the latter impose very strong priors on the structure of the synthesized images, which risks evolving exceedingly complex photorealistic details that are unlikely to be encoded by units in low- and mid-level processing stages (Wang & Ponce, 2024).

The CMA-ES optimizer is configured with the following hyperparameters:

- **Initial Step Size** ($\sigma_0$): 1.0
- **Population Size**: 50

The initial step size $\sigma_0$ determines the covariance of the sampling distribution at the onset of optimization, effectively controlling the exploration radius in the latent space.

The population size specifies the number of candidate solutions (i.e. codes) evaluated per iteration.

Initial codes were randomly sampled from a Gaussian distribution $\mathcal{N}(0, \sigma_{\text{init}})$. The initial standard deviation, $\sigma_{\text{init}}$, was set to $\sqrt{0.01 \times \mathbb{E}[|\boldsymbol{\xi}_{\text{ref}}|]}$, where $\mathbb{E}[|\boldsymbol{\xi}_{\text{ref}}|]$ is the mean absolute value of the reference code. For invariance experiments benchmarked against natural images, where direct reference codes were unavailable, initial codes were drawn from a standard normal distribution ($\mathcal{N}(0, 1)$).

## A.3 Pareto front computation

Multi-objective optimization often involves conflicting goals (e.g., stretching vs. squeezing, see Fig. 1). One way of dealing with such problems is by structuring the fitness function as a linear combination of the different optimization objectives (weighted-sum methods). However, these methods often require fine parameter tuning to prevent one objective from dominating. Unlike weighted-sum methods, Pareto fronts (PFs) handle trade-offs without parameter tuning. At each optimization iteration, candidate solutions (n=50) are ranked iteratively into PFs in the two-objective space. Each PF contains mutually non-dominant solutions (i.e. one is not better than another in both objectives). To get a full ordering, solutions are sorted within each front:

**Invariance Experiments:** In this scenario, solutions on the same Pareto front were prioritized based on their proximity to the target unit's reference activation level ($\min_x \| \Gamma(\boldsymbol{x}, \phi^\ell) - \boldsymbol{x}_{\text{ref}}^\ell \|_2$). This exploitative strategy was motivated by the empirical difficulty of converging towards the target activation level while at the same time maximizing input distance.

**Adversarial Attacks:** Solutions on the same Pareto front were selected for the next generation with uniform probability (random ordering). This promotes *exploration* along the front, preventing premature convergence to a single type of adversarial strategy.

CMA-ES then updates its distribution using the best $30\%$ of this ranking. At the end of the optimization, all solutions are used to compute a global PF, which is thus fully data-driven (see Fig. S1 for several examples). In our study, we analyzed only the last solution from this final PF for each experiment.

## A.4 Separability of the invariant images in the pixel representation

To assess whether the representation spaces where the stretching was applied yielded sets of invariant images that were distinguishable at the pixel level, we performed an image classification analysis. We generated invariant images for $n = 77$ readout layer units (i.e., 77 ImageNet classes) by stretching the representations in three different processing stages: (*i*) `low_level` (pixel space), (*ii*) `mid_level`, and (*iii*) `high_level`. For each unit and each representation space condition, we performed 10 independent optimization runs with different random seeds, yielding a total dataset of $77 \times 3 \times 10 = 2310$ images. Principal Component Analysis (PCA) was applied to the raw pixel representations of all images. The first $k$ principal components were then used as input features to train a Support Vector Classifier with a radial basis function kernel to discriminate the class to which the invariant images belonged (i.e., `low_level`, `mid_level`, or `high_level`). 5-fold cross-validation was used.

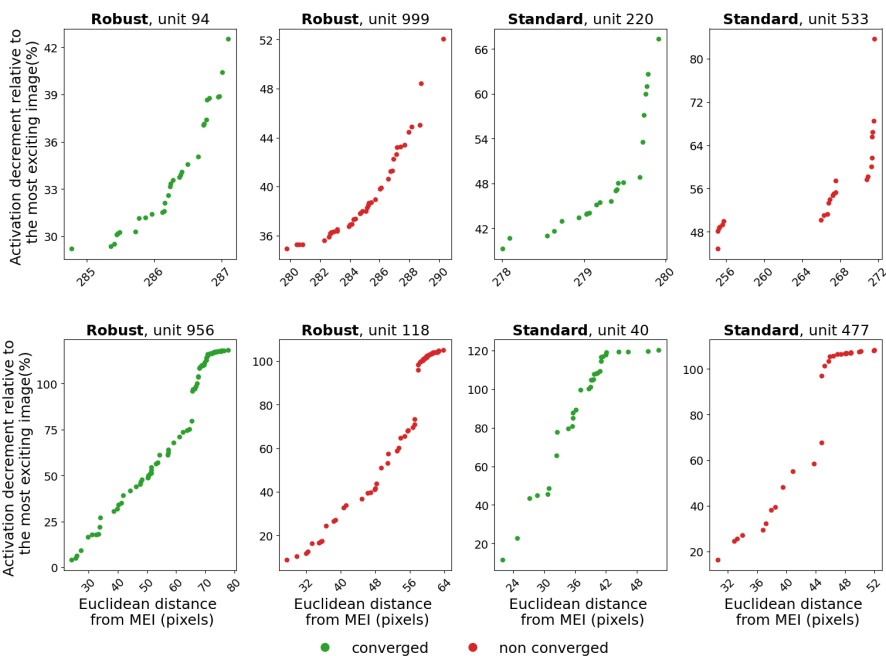

Figure S1: **Examples of solutions in the final Pareto Front.** Visualization of the final Pareto fronts for $n = 4$ example units in Invariance experiments (top) and $n = 4$ example units in Adversarial Attack experiments (bottom). In all reported experiments, targets were located in the readout layer, while pixel space was used as lower representation space. The ordinate axis represents the activation reduction of a readout unit of a robust or standard ResNet50 with respect to the activation produced by a reference image (MEI), when SnS is applied to synthesize either adversarial or invariant images, while the abscissa represents the $L_2$ pixel distance between these synthesized images and the reference. In the Invariance Experiments, SnS maintains a substantial fraction of the reference image activation (regardless of whether the optimization fully converges) while allowing the synthesized images to diverge markedly from the reference in pixel space. Conversely, in the Adversarial Attack setting, SnS reliably produces images that almost completely suppress the response of the target unit, yet remaining comparatively close to the MEI in the pixel space representation. Note the different scale range of the abscissa and ordinate axes between the top and bottom plots.

## A.5 OPTIMIZATION CONVERGENCE

### A.5.1 SNS IMAGES FALL INTO FUNCTIONALLY RELEVANT ACTIVATION REGIMES

Although some optimizations terminated at maximum iterations (adversarial: 97.92% (robust), 93.12% (standard); invariant: 63.38%, 35.71%, 21.30% (robust: `low_level`, `mid_level`, `high_level`); 86.75%, 32.60%, 5.84% (standard: `low_level`, `mid_level`, `high_level`)), final populations consistently reached functionally relevant activation regimes. Most invariant images achieved activations within the top 0.1% of natural image responses, being comparable or more extreme than the readout activation of images of the same category encoded by the readout target unit (Fig. S2a). Adversarial examples drove activations to the lowest $1^{st}$ percentile of the natural images, thus significantly suppressing target unit activation with respect to average activation with a natural image. As shown in Table S1, this was true also when considering only those units for which the early-stopping criterion was not reached, i.e., the optimizations that terminated at the maximum number of iterations (500). Moreover, as illustrated by the convergence curves in Fig. S2b, the average trajectories of both optimization objectives (activation of the target readout units and Euclidean distance from the MEI in pixelspace) rapidly approached stable asymptotic values consistent with the respective optimization goals (i.e., invariant and adversarial). In particular, the Euclidean distance plateaued, indicating that the optimized images no longer underwent substantial changes after a certain number of iterations.

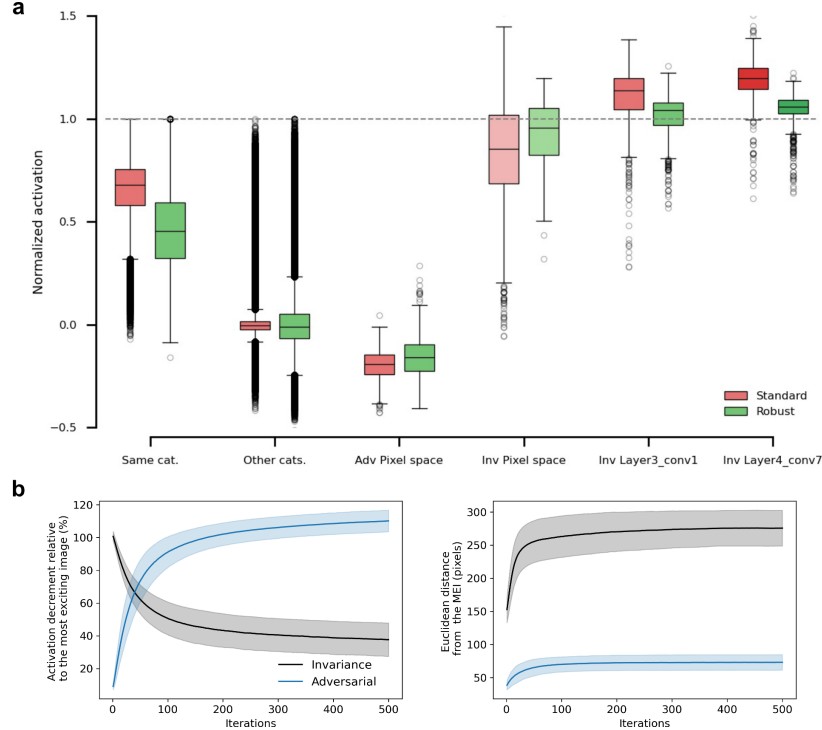

Figure S2: **SnS images fall into functionally relevant activation regimes.** **(a)** Distributions of readout target unit activations (normalized by peak natural stimulus response) for SnS-generated images: invariant (stretched at pixel, mid-, high-levels) and adversarial (pixel-space). Each distribution pulls together all optimizations performed in that specific condition (n = 77 target neurons x 10 random seeds = 770 experiments per condition). Comparison with natural image activations (same/different category) confirms that SnS invariant images elicit strong target responses, while adversarial examples achieve significant suppression. **(b)** The average convergence curves ($\pm$ SD) for the experiments illustrated in Fig. 2 (invariance and adversarial attacks, n = 770 runs per condition). Both optimization objectives (i.e., activation of the target units and Euclidean distance from the MEI in pixel space) are shown as a function of iteration number. Both objectives converged to stable asymptotic values consistent with the intended optimization target (i.e., the search for invariant or adversarial images). In particular, the Euclidean distance reached a plateau, indicating that the images no longer continued to evolve significantly following an initial period of convergence.

### A.5.2 INVARIANT SOLUTIONS ARE MAXIMALLY DISTINCT FROM THEIR REFERENCES AND HETEROGENEOUS ACROSS MULTIPLE RUNS

To quantify how much the invariant images generated by SnS (when targeting a given processing stage) diverged from the reference MEI ($x^\star$) across the depth of ResNet50, we computed the Euclidean distance between their activation vectors at each layer of the network (*Representational Distance Analysis* in short). Again, 10 different invariant images were produced for $n = 77$ distinct readout units by stretching representations at layers `low_level`, `mid_level`, and `high_level`. We calculated the mean ($\pm$ SEM) Euclidean distance between each of these $77 \times 10$ invariant images and the corresponding reference images at every convolutional, pooling and linear layer of the network, separately for each stretching stage. Analogously, to quantify heterogeneity between different invariant solutions, for each aforementioned stage we measured the average pairwise distance between invariant images, then averaged across each target neuron ($n = 77$). These distances were compared against three control metrics:

**Reference Variability distance:** Mean distance between all pairs drawn from 10 independent XDREAM runs for each unit ($n = \binom{10}{2} = 45$ pairs per unit, $n = 77$)
**Within-Category distance:** Mean distance between pairs of images randomly sampled from the same ImageNet category (mean over 1000 categories, 10 images/category).

Table S1: **Mean activations of target units to SnS-synthesized images from optimizations that failed to converge.** Reported activations are normalized by the response to the maximally activating natural image for the corresponding unit. **N** denotes the number of runs in which the early-stopping criterion was not reached for each experimental condition, out of a total of 770 optimization runs per condition. Successful maintenance of activation levels in Invariance runs would be indicated by positive activation values in the Mean column near to or exceeding 1, while successful suppression of activity in Adversarial runs would result in low or negative values.

| Experiment | Layer | Model Type | Mean | SEM | N |
|---|---|---|---|---|---|
| Invariance | low_level | Standard | 0.7682 | 0.0103 | 668 |
| Invariance | mid_level | Standard | 0.9581 | 0.0110 | 251 |
| Invariance | high_level | Standard | 0.9623 | 0.0233 | 45 |
| Adversarial | low_level | Standard | -0.1920 | 0.0027 | 718 |
| Invariance | low_level | Robust $L_2$ | 0.8467 | 0.0056 | 489 |
| Invariance | mid_level | Robust $L_2$ | 0.9104 | 0.0059 | 276 |
| Invariance | high_level | Robust $L_2$ | 0.9271 | 0.0075 | 166 |
| Adversarial | low_level | Robust $L_2$ | -0.1547 | 0.0036 | 755 |

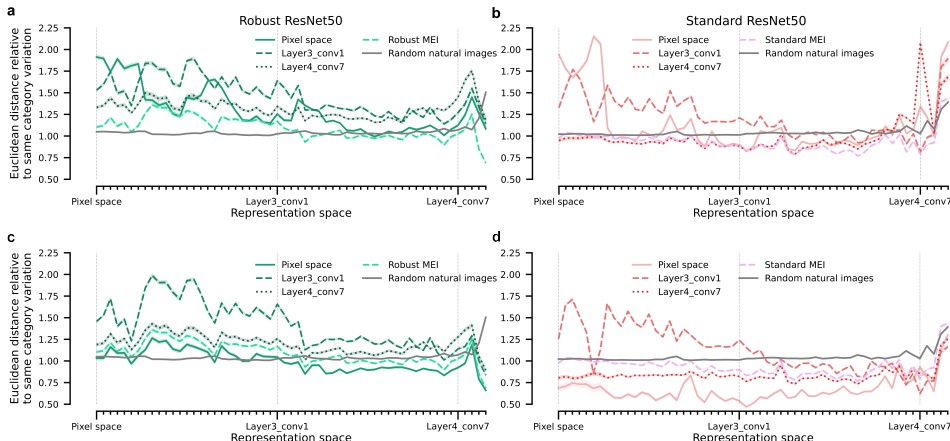

Figure S3: **Representation of the invariant images across the feedforward hierarchy.** (**a**) Average distance between the invariant images generated by SnS and their reference MEIs across the different stages of a robust ResNet50. Euclidean distances were normalized by the mean, within-category distance of Imagenet images. Different lines indicate the different stretching layers used in the optimization (also reported in the $x$ axis). The average distance between randomly selected natural images from Imagenet (solid gray line) and the average distance between multiple MEIs generated via XDREAM are reported for comparison. (**b**) Same as (**a**) for standardly trained network.(**c**) Normalized average distance between the invariant images generated by SnS across the different stages of a robust ResNet50. Different lines indicate the different stretching layers used in the optimization. The average distance between randomly selected natural images from Imagenet (solid gray line) and the average distance between multiple MEIs generated via XDREAM are reported for comparison. (**d**) Same as (**c**) for standardly trained network.

**Between-Category distance:** Mean distance between pairs of images randomly sampled across all ImageNet categories (using the same total number of pairs as the within-category control).

In our analysis (see Fig. S3) we normalized all Euclidean distances by the mean, within-category distance of Imagenet images. Fig. S3a-b show that invariant images that were generated to be maximally different from the MEI in a specific layer had the largest normalized distance in that layer, as compared to invariant images obtained by applying the stretching in other layers. Moreover, this distance consistently surpassed the one among natural images (both within the same category and randomly sampled) and multiple MEIs of the same unit, demonstrating SnS's efficacy in exploring larger spans of a unit's invariance manifold, as compared to searching multiple times for different

MEIs. Moreover, Fig. S3c-d show that distances between invariant images were significantly above 0 for all stretching layers (pixel space, mid, and late layers) in both standard and robust ResNet50, indicating a substantial diversity between invariant images. Heterogeneity is lowest in the pixel space (standard: 0.69, robust: 1.03) and higher in the mid (standard: 1.24, robust: 1.66) and late layers (standard: 0.85, robust: 1.27). Interestingly, invariances in the mid-layers of both networks - and those in the late layer of the robust network - are more heterogeneous than both random ImageNet pairs and multiple MEIs for the same unit. Instead, the invariant images in the late layer of the standard network exceed MEI heterogeneity, but not that of random ImageNet pairs. This heterogeneity is supported by SnS random initialization, which promotes broad exploration of the invariance landscape across multiple runs for the same unit and reduces the likelihood of converging to the same local minimum.

## A.6 COMPUTATIONAL RESOURCES

All experiments reported in this work were executed on a Dell Precision 7960 Tower (Ubuntu 22.04.5 LTS) with the following configuration:

- **CPU:** Intel (R) Xeon(R) w5-3425 (24 cores, 48 threads);
- **GPU:** NVIDIA RTX A6000 (48 GB GDDR6);
- **System Memory:** 128 GiB;
- **Storage:** 8.2 TB.

On this setup, a full optimization of a single SnS run (500 iterations) lasted approximately 120 seconds. The complete source code used to conduct the experiments is available at https://github.com/zoccolan-lab/SnS. Moreover the experimental data is accessible via Tausani et al. (2025).

## B PROCEDURE TO PROBE THE INTERPRETABILITY OF THE INVARIANT IMAGES BY HUMANS AND OBSERVER NETWORKS

- Thank you for your joining our study! This task involves making judgements about pictures.
- To start a trial, press the white "+" button at the center of the screen.
- Once the trial starts, a picture will be shown to you. Then, some choices will appear.
- **Your task** is to click the choice which best describes the first image.
- Making accurate choices will *increase your bonus payout*, but random guessing will lead to *no bonus* and to the task ending early.
- The experiment is about **40 minutes** long. Please feel free to take breaks in between trials as needed.
- Some trials may be extremely challenging or ambiguous. We cannot guarantee that it is possible to choose accurately in all cases.
- We reserve the right to end the experiment at any time.
- If you encounter a bug (e.g., the task freezing), please contact us and let us know. You will be compensated for your time.
- Data from your participation (e.g., which images you see, response times) may be included in publicly available datasets to support open science.
- Your personal information will remain confidential, though anonymized demographic information may be included in summary statistics.
- If you wish, you may view our full consent form document here.
- By clicking "Continue," you voluntarily agree to be a participant in our experiment and agree to all of the rules above.

Continue

Figure S4: **Instructions shown to human participants**. These instructions were shown to the participants at the beginning of the task.

For our interpretability experiments, we chose 12 ImageNet categories (`goldfish`, `chickadee`, `frilled lizard`, `Dungeness crab`, `Staffordshire bull terrier`, `cicada`, `candle`, `grand piano`, `minibus`, `reflex camera`, `soccer ball`, `cup`). For each category, we selected 10 reference images $x_{\text{ref}}$ and, for each of them, we generated one invariant image by stretching its representation in `low_level`, `mid_level`, and `high_level` in both standard and robust ResNet50. This yielded 6 kinds of invariant images per reference image, where invariance was always defined relative to the readout unit corresponding to the reference image's category. For each category, we also included in the classification pool the 10 reference images and 20 MEIs (generated with XDREAM) of the readout unit corresponding to the chosen category (10 for the standard and 10

for the robust network). This yielded a total of 12 categories $\times$ 10 randomly sampled images $\times$ 9 stimulus types (i.e., 1 reference image, 2 MEIs and 6 invariant images). Observer networks had to classify all 1080 of these images; humans had to classify a subset of 540 images (i.e., 5 rather than 10 randomly sampled images per category). Classification accuracy (12-AFC percent correct) was calculated for each of the 9 stimulus types. To assess the generalizability of our findings, we repeated the same experiment with invariant images generated by two additional architectures (ResNet18 and VGG16_bn, in both standard and robust versions; see details above). The pool of observer networks was identical to the previous experiment. As for ResNet50, 3 levels of stretching were applied: Pixel space, mid level (VGG16_bn: $2^{nd}$ convolutional layer in the $3^{rd}$ block; ResNet-18: $4^{th}$ convolutional stage in layer 2) and high level (VGG16_bn: $3^{rd}$ convolutional layer in the $5^{th}$ block; ResNet-18: $4^{th}$ convolutional stage in layer 4).

The psychophysics experiment administered to human participants was conducted as follows. After clicking on a fixation cross at the start of each trial, participants were presented with an image for 50 ms at $\sim 10°$ of visual angle. After a 20ms blank screen, a backward mask image (random RGB pixel noise) was then presented for 200ms. Images were presented in random order. Participants then clicked one of 12 category buttons that subsequently appeared in a circular arrangement, with a 10-second timeout (Fig. 4a). To avoid directional bias, the sequence of category buttons was randomly rotated for each trial.

Participants for the human experiment were recruited using the online Prolific platform under a preexisting IRB protocol. Participants were compensated at an overall rate calibrated to \$15 USD per hour. To facilitate engagement with the task, compensation included a 1-cent bonus for each correct trial, and participants were shown a green check after correct responses or a black "X" otherwise. The typical bonus amount that participants would receive was overestimated based on pilot data. To adjust for this discrepancy, after recruitment for the study was complete, participants were uniformly provided with an additional bonus to bring compensation to the \$15/hour level.

The main task was preceded by a screening phase, in which participants classified 24 natural images (2 per category), and were allowed to proceed if they correctly classified $\geq 1$ image per category and $\geq 16$ in total (maximum 3 attempts allowed per participant). Data from the screening phase were not included in any analyses. Participants who did not pass the screening phase were compensated for the time spent during the screening process.

The experiment posed minimal risks to participants. It is unlikely but conceivable that the brief presentation of images and masks could be hazardous to a subset of individuals with photosensitive epilepsy: out of an abundance of caution, a warning was shown in the description of the task on the Prolific website to be viewed by participants before joining the study. To ensure consistent stimulus presentation, participants were required to complete the task on a desktop or laptop computer; access via tablets and smartphones was programmatically blocked. The title and description of the task on the Prolific website are reproduced as follows:

**Title:** *Identify objects in photos, earn roughly \$4.00 bonus for accurate responses*

**Description:**

*View 540 photos and identify the animal or other object in each photo. Earn a bonus for each accurate response, typically around \$4.00 USD in total (maximum \$5.40).*

*PLEASE NOTE:*

1. *The task begins with a screening phase you must pass with a certain accuracy level before starting the experiment proper. You may re-attempt the screening phase up to 2 times if you wish, but you will only be compensated for a maximum of 5 minutes spent in the screening phase.*

2. *Please avoid resizing the task window during the experiment, as this can break the task. It's best to maximize your browser window and get comfortable before starting.*

**WARNING:** *this task contains bright, rapidly flashing images: it is not suitable for individuals with photosensitive epilepsy.*

Please see Fig. S4 for the additional in-task instructions provided to participants. The task was implemented using the JsPsych library (De Leeuw, 2015) and the JsPsychPsychophysics plugin (Kuroki, 2021).

Table S2: **Statistical results from omnibus test (type II Wald chi-square) and post-hoc pairwise comparisons to test differences among stimulus types in the human image classification experiments**. All $p$ values are adjusted using the Benjamini-Hochberg correction, with significant values in bold.

| | Human subjects | |
| --- | --- | --- |
| **Comparison** | **z-value** | **p-value** |
| Omnibus test (effect of stimulus type) | - | **<0.0001** |
| Average Robust Stretch vs Average Standard Stretch | 36.63 | **<0.0001** |
| Robust layer4_conv7 vs Robust layer3_conv1 | -4.94 | **<0.0001** |
| Standard layer4_conv7 vs Standard layer3_conv1 | -8.41 | **<0.0001** |
| Robust layer4_conv7 vs Robust pixel space | -9.52 | **<0.0001** |
| Standard layer4_conv7 vs Standard pixel space | 7.49 | **<0.0001** |
| Robust layer3_conv1 vs Robust pixel space | -4.66 | **<0.0001** |
| Standard layer3_conv1 vs Standard pixel space | -0.99 | 0.32 |
| Robust MEI vs Standard MEI | 32.96 | **<0.0001** |

## B.1 STATISTICAL ANALYSIS FOR CLASSIFICATION EXPERIMENTS

Analysis of data from classification experiments focused on identifying accuracy differences between 9 image types: natural images, MEI's from robust and standard networks, and invariant images from 3 different layers for both network types. We fitted a generalized linear mixed model (GLMM) with a logistic link function for binary correct vs. incorrect trial responses, including random intercepts for participants and semantic image categories. We used an omnibus test (type II Wald chi-square) to assess overall differences among the 9 image types, followed by 8 planned contrasts via estimated marginal means: (i) robust vs. standard MEIs, (ii) robust vs. standard invariant images averaged across the 3 representation layers, and (iii-viii) within each network type, all 3 possible pairwise comparisons between invariant types at different layers. Our analysis employed the Benjamini-Hochberg procedure to control the false discovery rate under multiple comparisons (Benjamini & Hochberg, 1995). The results of the planned comparisons are provided in Table S2.

## C SnS GENERATES EFFECTIVE ADVERSARIAL AND INVARIANT EXAMPLES FOR NATURAL IMAGES

We replicated the experiment reported in Section 4.1 (see Fig. 2) also in the case where, for a target readout neuron, a highly effective natural image representative of the class encoded by the unit was used as reference for SnS optimization instead of a MEI. As for Section 4.1, both invariant and adversarial images were synthesized by stretching the pixel-level representation. In these experiment, we considered the 12 readout units of robust ResNet50 encoding the 12 image categories used for the interpretability experiment with human observers (see Sections 3.2, 4.3 and Supplementary Section B). For each unit, both the adversarial and invariant experiment was repeated for 10 different reference natural images, the same ones used for the interpretability experiment (for a total of $12 \times 10$ invariant images and $12 \times 10$ adversarial images). Example images are shown in Fig. S5b, bottom. Fig. S5a shows that, similarly to the experiment performed with MEI references (see Fig. 2), both adversarial and invariant images reached the desired functional activation regimes (i.e., for adversarial images: silencing the target unit without distancing too much from the reference image; for invariant images: departing substantially from the reference natural image, while at the same time preserving the same level of activation).

## D SINGLE-LOSS ABLATION EXPERIMENTS

To assess the functional contribution of each component of the SnS dual-loss objective (see Section 3.1 for a detailed description), we performed a set of ablation experiments in which we optimized images using only one loss term at a time. We repeated the experiment shown in Fig. 2, in which a readout unit is selected as the optimization target and pixel space is used as the representation

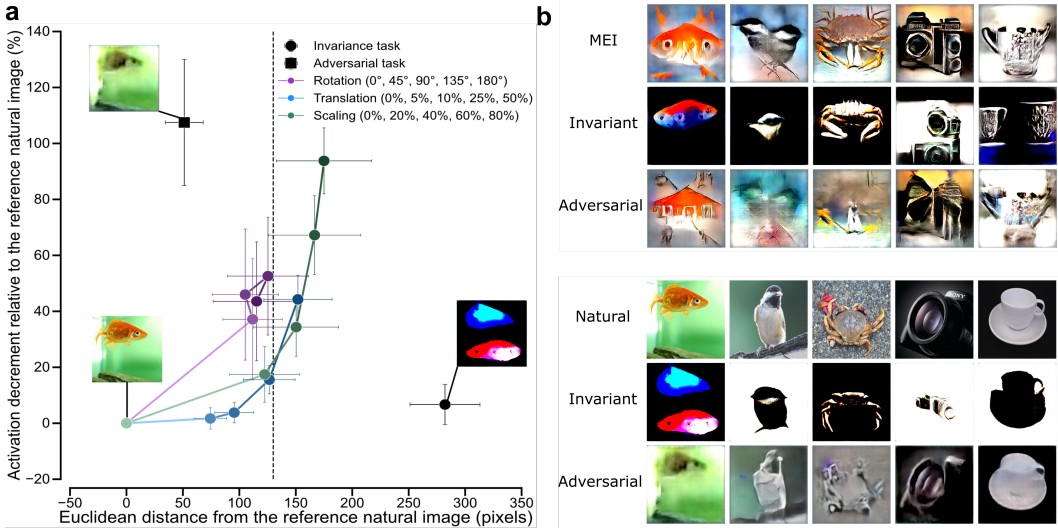

Figure S5: **SnS generates effective adversarial and invariant examples for natural images** (**a**) As in Fig. 2, the ordinate axis represents the activation reduction of a readout unit of $L_2$-robust ResNet50 with respect to the activation produced by a reference natural image, when SnS is applied to synthesize either adversarial or invariant images, while the abscissa represents the $L_2$ pixel distance between these synthetized images and the reference. In this experiment, differently from Fig. 2, the reference images were not the MEIs of the readout units but natural images belonging to the specific categories encoded by the units. Each data point reports the average ($\pm$ SD) of these metrics across $12 \times 10$ image synthesis experiments, where 12 are the targeted readout units and 10 are the different reference images selected for each unit. (**b**) Examples of invariant and adversarial images obtained for 5 readout units of $L_2$-robust ResNet50, when the reference images were either their MEIs (top), as for the experiment shown in Fig. 2, or natural images (bottom), as for the experiment shown in (**a**).

space. For each target unit, we synthesized images under four single-loss variants: 1) Squeeze-up: minimize the target unit activation distance between the synthesized image and the reference MEI; 2) Stretch-up: maximize the activation distance from the reference MEI; 3) Squeeze-down: minimize the distance of the synthesized image from the reference MEI in the pixel representation; and 4) Stretch down: maximize such distance.

We used the same 77 readout units and 10 initialization seeds per unit as in the main experiments, resulting in 770 runs per single-loss condition. Squeeze-up and stretch-down followed the initialization and Pareto-front sorting used for SnS invariances; squeeze-down and stretch-up followed the procedures used for SnS adversarial optimizations (see Sections A.3 and 3.1).

Across all four variants, optimizing only one objective consistently failed to drive the other metric (activation distance or pixelspace distance) to its required regime (Fig. S6). None of the single-loss optimizations produced solutions that simultaneously exhibit the trade-off characteristic of SnS invariant or adversarial synthesis. This demonstrates that the dual-loss formulation is necessary for producing meaningful invariances and adversarial examples. Although the single-loss variants did not reproduce SnS behavior, each produced a characteristic synthesis regime that sheds light on the role of its loss component:

- Squeeze-up: Optimizing only the squeeze term yielded images whose activation approached the invariant regime (i.e., close to the MEI's activation), but whose pixel-space distance remained near the distance of the initial random seeds (see the black curve in Fig S2b, right when iterations is equal to 0). This behavior mirrors repeated XDREAM optimization with multiple initializations, producing multiple high-activation images that differ from each other but do not explore the far boundary of the invariance manifold. The explicit maximization of representation distance (e.g., in pixel space) is the factor that drives invariant images far

away from its reference MEI, exploring the maximal invariant transformations within that representation space.

- Stretch-up: Maximizing the activation distance naturally pushed images to diverge in pixel space as well. The optimization suppressed the target unit by making the image increasingly dissimilar from the reference. This behavior is essentially the opposite of SnS adversarial synthesis, where the unit is silenced with minimal pixel-space change.

- Squeeze-down: Matching the reference MEI in pixel space also preserved the high activation of the reference. This optimization is notably similar to that used for model metamers (Feather et al., 2023), which enforce representation matching at internal layers. Thus, metamer synthesis emerges as a special case of the SnS loss structure when the squeeze term is applied alone to the representation space.

- Stretch-down: Maximizing pixelspace distance alone caused images to diverge visually but produces no systematic change in target-unit activation. Indeed the activation of the target unit remained close to its initial value at the beginning of the optimization (see the black curve in Fig. S2b, left; iteration = 0), which is very far from the one produced by the MEI. This confirms that stretch-down alone does not find images relevant to the unit of interest.

In summary, the single-loss ablations showed that none of the individual terms is sufficient to generate the invariant or adversarial images produced by SnS. The dual-loss structure is essential, with each component stabilizing and constraining the other. These results further validated SnS as a principled framework for probing model invariances and adversarial directions.

# E  SnS IS EFFECTIVE IN TARGETING UNITS IN HIDDEN LAYERS

Building upon our findings in Section 4.1, we extended the application of SnS to units within hidden convolutional layers. Unlike output layer units tuned to predefined classes, hidden units are hypothesized to function more analogously to biological visual neurons, responding to intermediate-level features rather than complete objects. This makes them compelling targets for deciphering learned internal representations.

To investigate this, we conducted SnS invariance experiments targeting 50 distinct units in both `layer3_conv1` and `layer4_conv7` of the ResNet50 network, using pixel space as the input representation. For each unit, we performed 10 optimization runs initiated with different random seeds. In the standard network, SnS terminated by early stopping in 67.20% of runs at `layer3_conv1` and 27.40% at `layer4_conv7`; in the $L_2$-robust network, convergence was 87.80% and 89.40%, respectively. Importantly, even in non-converging instances, the optimization process often guided the input towards stimuli that elicited high unit activations, consistent with functionally relevant regimes identified through natural image statistics on ImageNet (Fig. S7e).

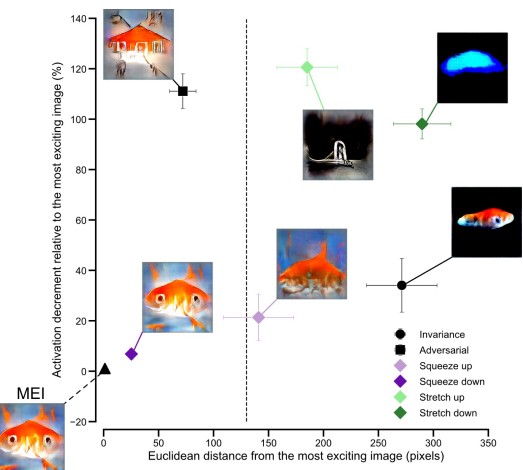

Figure S6: **Single-loss ablation experiments demonstrate the need of SnS dual loss.** Comparison between SnS adversarial and invariance experiment distances (same as in Fig. 2) with the results of single loss optimizations (squeeze up, squeeze down, stretch up, stretch down). Each point corresponds to the mean ± SD of 770 experiments (i.e. 77 readout units x 10 random initializations per each unit). The black dotted line represents the median image $L_2$ distance among Imagenet images.

Qualitatively, the SnS-generated invariant images served as effective "feature visualizers." This was particularly insightful as the precise selectivity of these hidden units is unknown a priori. The

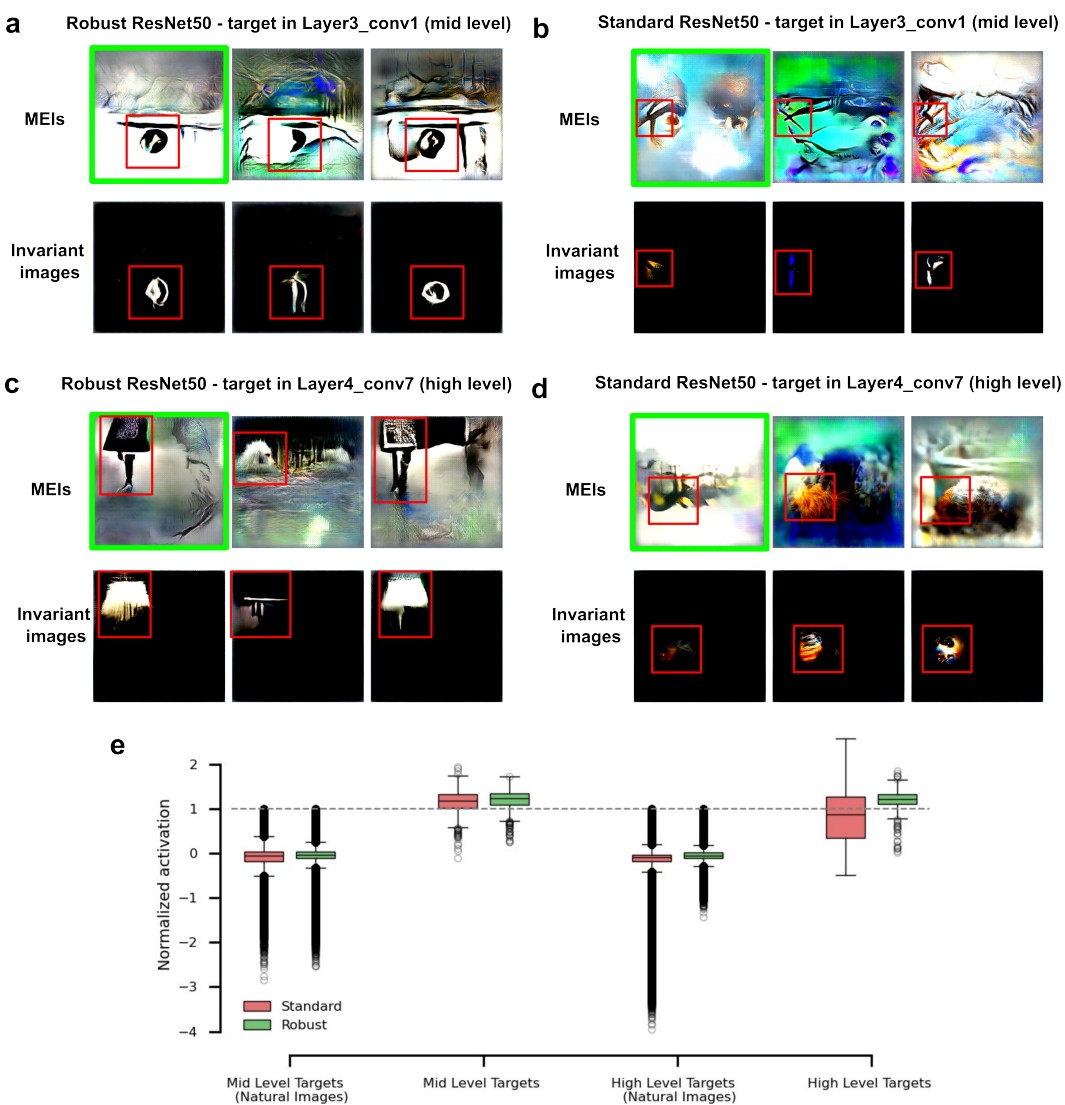

Figure S7: **SnS is effective with targets in hidden layers** (**a**) Comparison of MEIs (top row) with SnS-generated invariant images (bottom row) for a representative target unit in `Layer3_conv1` of robust ResNet50. SnS images, synthesized to be invariant in pixel space, were generated using the green-highlighted MEI as a reference. Red rectangles identify the unit's key responsive regions. This comparison highlights SnS's ability to distill the core visual features a unit detects, offering clearer insight than MEIs alone. (**b-d**) The same SnS-based feature visualization approach (as in (**a**)) is applied to different representative target units: (**b**) `Layer3_conv1` of a standard ResNet50; (**c**) `Layer4_conv7` of a robust ResNet50; and (**d**) `Layer4_conv7` of a standard ResNet50. (**e**) Final activation distributions for the target units from the SnS optimization in hidden layers. As in Fig. S2a, convergence performance is compared to natural image statistics

resulting visualizations clearly delineated the specific image features to which a target neuron was most sensitive (region of interest (ROI)), effectively isolating these core features from irrelevant contextual details (Fig. S7a-d (bottom rows)). This level of feature disentanglement and clarity surpassed that observed in images optimized by methods like XDREAM (Xiao & Kreiman, 2020), which often retain more complex, less isolated visual elements (Fig. S7a-d (top rows)). Moreover, by comparing the reference image to multiple invariant instances (i.e. initialized with multiple random

seeds), the images revealed specific transformations the identified ROI could tolerate while preserving the unit's activation level, offering insights into the unit's invariances (Fig. S7a-d (bottom rows)).

## F  SNS CAN BE USED TO STUDY INVARIANCE MANIFOLD GEOMETRY

SnS can be used to extensively sample the invariant manifold of a target unit, paving the way to the analyses of its intrinsic dimension (ID) and geometric structure. To show the feasibility of this approach we selected a single target readout neuron (the `chickadee` unit in the $L_2$ robust ResNet50 readout layer) and extensively sampled its maximally invariant images across three different stretching layers (the same ones used in Section 4.2): pixel space (input/low-level), `Layer3_conv1` (mid-level), and `Layer4_conv7` (high-level). For each stretching layer, we ran SnS 100 times for each of 5 different reference MEIs (obtained from independent XDREAM runs), yielding 500 invariant images per layer. To estimate the ID of each invariance manifold, we applied two distinct techniques to the vector representations of the invariant images: 1) PCA, measuring the number of components explaining 95% of the variance; and 2) two-nearest-neighbors intrinsic dimension estimation (2NN-ID; Facco et al. (2017)), a state-of-the-art, nonlinear estimator of intrinsic dimension that has been successfully applied to analyze image manifolds in both CNNs and visual cortical areas (Ansuini et al., 2019; Muratore et al., 2022a). 2NN-ID was computed for different subsamples of points (from $n = 50$ to $n = 500$ invariant images), repeating the process 50 times for each cardinality.

These two approaches returned dimensionalities in very different numerical ranges: PCA required hundreds of components to explain 95% of variance (from 230 to 340 PCs, Fig. S8a), whereas the ID estimated by 2NN-ID ranged between 10 and 30 across the three manifolds (Fig. S8b). This radical difference in ID estimation between a classic linear method (PCA) and 2NN-ID points to the fact that the structures of all 3 manifolds are highly nonlinear (Facco et al., 2017; Ansuini et al., 2019). This aligns with expectations for neurons high in the visual hierarchy whose responses abstract over complex real-world transformations.

Despite differences in scale, both methods agreed on a consistent ordering of dimensionality across stretching layers: lowest ID for pixel-level stretching, highest for mid-level stretching, and reduced ID again for high-level stretching. This pattern mirrors previously reported trends in the ID of image representations across deep CNNs (Ansuini et al., 2019), where ID initially increases, peaks in mid layers and then gradually diminishes in deeper layers. This result further reinforces the distinction between invariances uncovered at different stretching layers (Fig. 3), indicating the ability of SnS to discover axes of variation that are distinct not only in qualitative terms (Fig. 3a), but also in their dimensionality.

## G  SNS IS ROBUST TO SUBSAMPLING OF THE REPRESENTATION SPACE

SnS, as a gradientless and model-agnostic method, offers significant advantages for neuroscience research, particularly in scenarios where constructing detailed "digital twin" models (as described in Section 2) is impractical. However, a critical challenge for applying SnS to study hierarchical invariances in the brain is the inherent undersampling of representational spaces. Despite recent advances in electrophysiology and calcium imaging enabling simultaneous recording from hundreds or thousands of neurons (Jun et al., 2017; Sofroniew et al., 2016), these recordings invariably capture only a fraction of the total neural population within a given brain area.

To evaluate SnS's performance under conditions analogous to experimental neuroscience, we investigated how subsampling the representational space impacts the qualitative nature and quantitative characteristics (assessed via representational distance analysis, see Section A.5.2 for methodological details) of the invariances identified by SnS. Specifically, we focused on the same 77 readout target units previously analyzed in Section 4.2. For these units, we identified invariances after subsampling their corresponding representational spaces (either `layer3_conv1` or `layer4_conv7` of the CNN) to either 1000 or 100 randomly selected units. To ensure robustness, this process was repeated 10 times for each target unit and subsampling condition, using different random selections of representational space units. As a reference, 1000 units represent 0.5% of the units in `layer3_conv1` ($n = 200704$), while they constitute 4% of `layer4_conv7` ($n = 25088$).

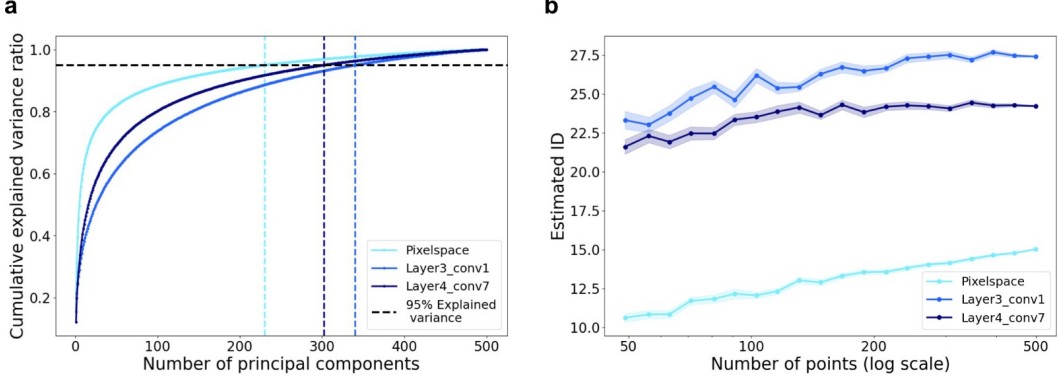

Figure S8: **Hierarchical invariance manifolds differ consistently in their dimensionality**. (**a**) PCA estimation of the dimensionality of the invariance manifolds obtained for an example $L_2$ robust ResNet50 readout unit by stretching the representations of the MEIs at different depths of the network: low-level (pixel space), mid-level (`Layer3_conv1`) and high-level (`Layer4_conv7`). The curves show the cumulative fraction of variance of each manifold (consisting of 500 sampled images) that was explained by considering an increasingly larger number of principal components (ranked according to their eigenvalue). For each stretching depth, the number of principal components necessary to explain 95% of data variance (horizontal black dashed line) is indicated by a color-specific vertical dashed line. (**b**) Dimensionality of the same invariance manifolds analyzed in **a**, but estimated using the nonlinear intrinsic dimension method 2NN-ID. The curves show the average intrinsic dimension $\pm$ SEM, as obtained for increasingly larger sets of invariant images sampled from the manifolds.

Our findings indicate that SnS is robust to such undersampling. Qualitatively, the invariances identified using subsampled spaces were comparable in their level of abstraction to those found using the full representational layer (Fig. S9a and c). This suggests that the SnS optimization process effectively identified relevant axes of image variation even with reduced unit information. This qualitative observation was corroborated by quantitative representation distance analysis (Fig. S9b and d), which demonstrated a strong alignment in the progression of invariance discovery between analyses using full and subsampled layers. These results highlight SnS's potential for reliably characterizing neural invariances even when constrained by the partial observations typical of in vivo recordings.

A plausible explanation for this robustness to subsampling lies in the inherent representational redundancy present in both biological neural circuits and artificial neural networks. In biological systems, information is often encoded in a distributed manner across populations (Zohary et al., 1994) (Stringer et al., 2019), where multiple neurons may carry similar or overlapping information, contributing to robustness against noise and cell loss (Pouget et al., 2000). Similarly, deep neural networks, including CNNs, have been shown to learn redundant features or possess over-parameterization, where multiple units or weights contribute to similar computations (Denil et al., 2013). Consequently, a sufficiently large and representative subsample of units can still capture the dominant computational characteristics and drive the overall representation into similar regimes as the full layer, explaining the observed consistency in identified invariances.

## H SnS DISCOVERS LAYER SPECIFIC HIERARCHICAL INVARIANCES FOR $L_\infty$-ROBUST MODELS

To assess whether hierarchical divergence in SnS invariances also arises under $L_\infty$ robustification, we generated SnS invariances for an $L_\infty$-robust ResNet50 ($\epsilon = 4/255$) from RobustBench (Croce et al., 2020) using the same n = 120 natural images and the same three stretching depths (pixel-space, mid-level, high-level) as in Fig. 4. Qualitatively (Fig. S10a) and quantitatively (Fig. S10b), layer-specific invariances in the $L_\infty$-robust model remained highly distinct and discriminable in pixel space, mirroring those already shown in Fig. 3. We then tested interpretability across observers by presenting

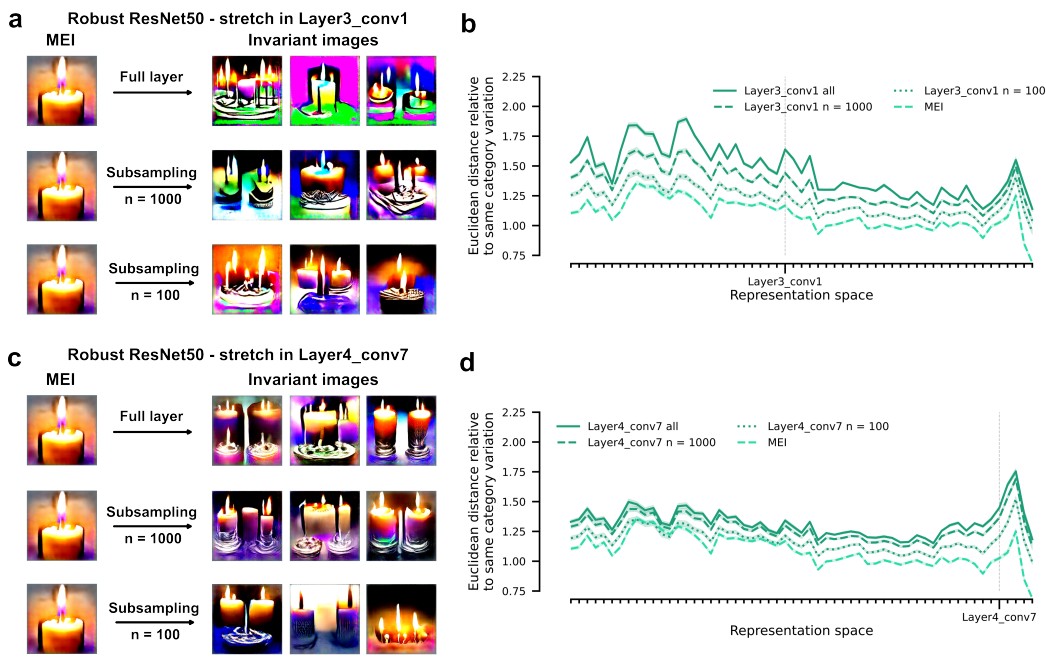

Figure S9: **SnS invariance is robust to subsampling of the representation space** (**a**) Example MEI and associated invariant images obtained for a readout neuron (`candle`) in $L_2$ robust ResNet50 stretching the mid_level layer (i.e. `Layer3_conv1`). Each row represents examples for different levels of subsampling: using the full layer ($1^{st}$ row), subsampled spaces of size $n = 1000$ and $n = 100$ ($2^{nd}$ and $3^{rd}$ rows respectively). Multiple results are shown for different random initialization seeds (columns). (**b**) Normalized average distance between the invariant images generated by SnS (stretching in `Layer3_conv1`) and their reference MEIs across the different stages of a robust ResNet50. Different lines indicate different cardinalities of units in the representation space used for optimization (i.e. all, $n = 1000$ and $n = 100$). The average distance between multiple MEIs generated via XDREAM is reported for comparison. (**c-d**) Same as (**a**) and (**b**) respectively, but stretching was performed on robust ResNet50 in the high_level layer (i.e. `Layer4_conv7`).

the $L_\infty$-robust invariances to three architectures (ResNet-18, Wide-ResNet50-2, ConvNeXt-Base), each in standard (Paszke et al., 2019) and $L_\infty$-robust versions ($\epsilon = 4/255$, also from RobustBench (Croce et al., 2020)). As in the $L_2$ setting, invariances from the $L_\infty$-robust generator were more interpretable on average than those from a standard ResNet50, and interpretability again decreased when robust invariances were evaluated by standard networks (Fig. S10c). Notably, however, when $L_\infty$ invariances were judged by other $L_\infty$-robust observers, classification accuracy either remained at ceiling or increased across stretching depths, suggesting that $L_\infty$-robust networks share a highly aligned invariance structure even at deeper layers. This observation highlights promising directions for future work comparing invariances across robustness regimes, as well as in models whose robustness arises from perceptually aligned training strategies (e.g., harmonized neural networks (Fel et al., 2022)) or biologically inspired architectures such as VOneResNet (Dapello et al., 2020).

## I    HIERARCHICAL INVARIANCES IN VISION TRANSFORMERS

One of the strengths of SnS is that, being gradient-free, it can easily be applied (literally out-of-the-box) to virtually any visual processing architecture (biological or artificial), as long as one can measure the responses of its units to the images produced by SnS optimization algorithm. In cases where the goal is to uncover the invariance of the whole model to transformations of specific object classes, the additional constraint is to have access to the responses of the readout units for those

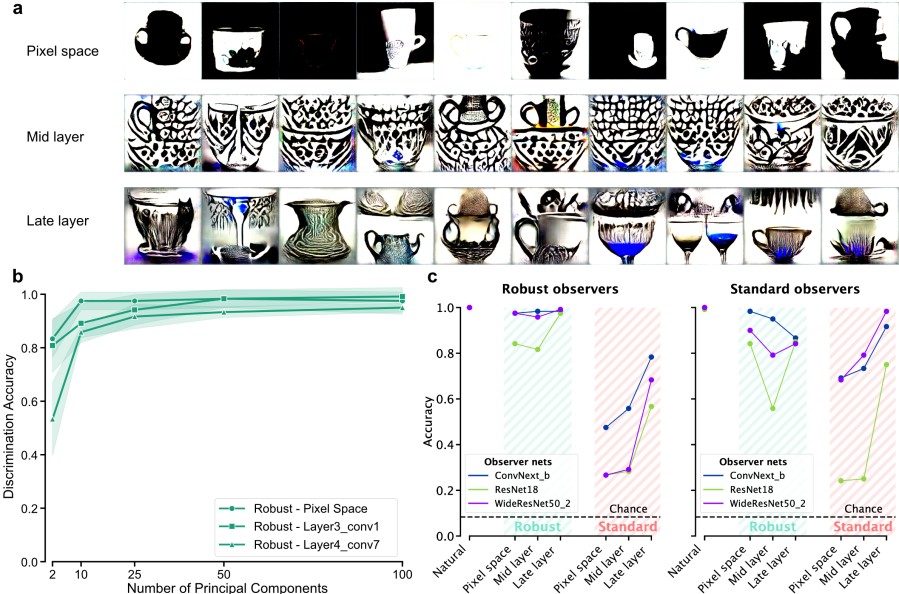

Figure S10: **Invariances in $L_\infty$-robust ResNet50.** (**a**) Example invariant images obtained for a read-out neuron ("cup") in $L_\infty$-robust ResNet50 and computed for `Pixel space`, `Layer3_conv1`, `Layer4_conv7` (rows). (**b**) Accuracy of a SVC in discriminating the three classes of invariant images produced by stretching at the aforementioned layers as a function of the number of principal components fed to the classifier. (**c**) Classification accuracies for $L_\infty$-robust and standard ResNet50 invariances across multiple $L_\infty$-robust and standard observer networks.

classes. As such, SnS can directly be applied to investigate the invariance of readout units in vision transformers (ViTs).

To test this application, we ran the experiments of Section 3.2 (observer networks only) on a ViT-B-16 model pretrained on ImageNet and a CLIP-trained ViT-B-16 visual encoder (Radford et al., 2021). Since the CLIP model lacks explicit readout units, we trained a linear classifier atop the frozen visual encoder for ImageNet classification to construct class-specific detectors (top 1 validation accuracy: $78.94\%$). For both models, we considered the output linear layer of the MLP in the sixth encoder layer as mid-level layer and the one in the twelfth encoder layer as high level representation.

Our findings reveal that, similar to CNNs, the nature of the discovered invariances in ViTs depends on the hierarchical level of the stretched representation. Stretching the pixel space primarily altered low-level properties such as luminance, yielding invariances that were poorly interpretable (Fig. S11 a,b). Contrary to the experiments with ResNet50 (see Fig. 3 a,b), where a clear, qualitative distinction was observable between invariant images stretched in mid and late layers, in both ViTs we did not observe such difference (see Fig. S11 a,b). These images were also comparable in terms of invariance interpretability (Fig. S11 c). This observation is consistent with the view that ViTs learn less strictly hierarchical representations than CNNs, and that these models integrate information more globally (Dosovitskiy et al., 2020). Overall, there seems to be a little, but consistent across conditions advantage of the CLIP-trained version of ViT-B-16 in terms of interpretability. This trend seems to be similar between standard and robust network observers.

## J SnS INVARIANCES AND METAMERS ARE STRUCTURALLY AND FUNCTIONALLY DISTINCT

To better contextualize SnS invariances within state-of-the-art techniques to probe invariances in neural networks, we carried out a systematic comparison between SnS-derived invariant images and gradient-based metamers. Metamers were generated using the original implementation of Feather et al. (2023), including the same hyperparameter settings mentioned in the original paper. Metamers

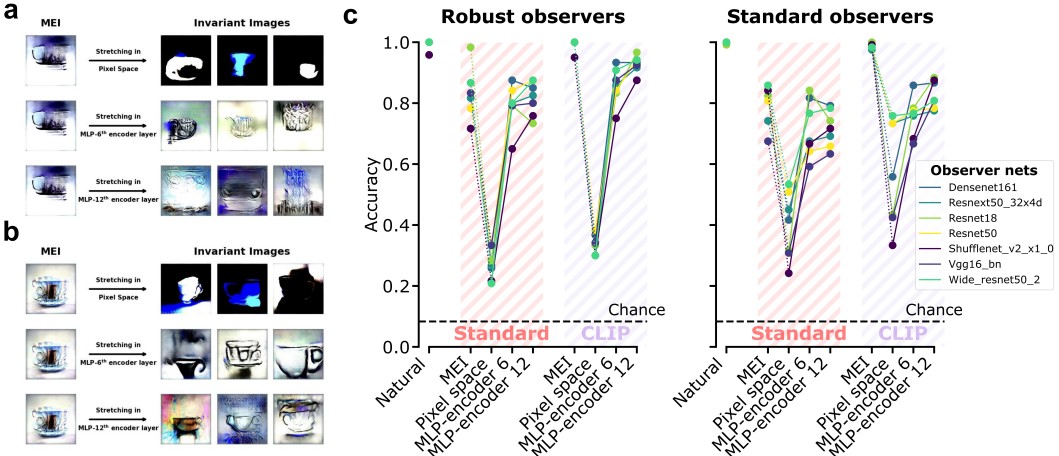

Figure S11: **Invariances in vision transformers**. (**a**) Example MEI and associated invariant images obtained for a readout neuron (`cup`) in standard ViT and computed for three different choices of the stretching representational stages (rows). Multiple results are shown for different random initialization seeds (columns) (**b**) Same as (**a**) but for a ViT encoder trained via CLIP. (**c**) Classification accuracies for ViT invariances across multiple robust (left) and standard (right) networks.

were synthesized for the same $n = 120$ natural images used in our behavioral experiment (Fig. 4), yielding for each natural image one SnS invariant and one metamer at each of three representational depths (pixel space, mid-level, high-level) in both standard and $L_2$-robust ResNet50 (for details see Section 3.1). First we compared metamers and SnS invariances in terms of interpretability. To do this, we replicated the experiment described in Fig. 4c with metamer stimuli. Using the same observer networks as in our main experiments (six architectures, each in both standard and $L_2$-robust versions; $n = 12$ total (Section 4.3)), we recovered results consistent with Feather et al. (2023) (Table S3). Specifically, metamers produced by the $L_2$-robust ResNet50 remained highly interpretable across observer models at all synthesis depths (pixel space, layer 3, layer 4). In contrast, standard-network metamers showed the expected sharp decline in interpretability for late-layer metamers (layer 4) across all observer networks. Critically, this pattern is the opposite of what we observed for SnS hierarchical invariances (Fig. 4c; numerical values in Table S3). Indeed, for standard networks, late-layer SnS invariances show a pronounced increase in interpretability, so much so that observer accuracies far exceed those obtained for late-layer metamers for the standard network.

A second point of stark contrast between the two methods is that of average distances from the reference natural images from which both metamers and SnS invariant images were generated (Table S4). Indeed, metamers are explicitly optimized to minimize representational distance to the target image at a given layer, whereas SnS invariances explicitly maximize that distance. As a result, metamers lie much closer to their reference images than SnS invariances at all depths and for both standard and robust generator networks. For comparison, we also report the average pairwise distances between natural ImageNet images from the same class at the same three layers (i.e., within-category distance for the three representation stages, see Supplementary Section A.5.2). While metamers lie far below these natural-image baselines, SnS invariances systematically exceed them by a wide margin (Table S4). In the context of the different image synthesis between the two methods, we remind that, as detailed in Supplementary Section D, metamer-like stimuli arise as a special case of the SnS objective (i.e. the squeeze-down single loss variant in Fig. S6). In this sense, SnS provides a gradient-free generalization of metamer-like synthesis, enabling analogous invariance characterizations in settings - such as in vivo experiments - where gradients are not accessible.

Table S3: **Interpretability comparison between metamers and SnS hierarchical invariances**. Classification accuracies (mean $\pm$ SEM) of the standard and $L_2$-robust observer networks used in the interpretability experiment (Fig. 4c), evaluated on images generated either as metamers or using SnS with both standard and $L_2$-robust ResNet50 as the generator network (**Gen Net** column). Metrics are reported for the three different representation layers used also in Fig. 4 (**Net Layer** column). Note that accuracy results for the metamers conditions are in line with Feather et al. (2023).

| Condition | Gen Net | Net Layer | Standard observers | Robust observers |
|---|---|---|---|---|
| Metamer | Standard | Pixel space | $0.999 \pm 0.00122$ | $0.989 \pm 0.00653$ |
| Metamer | Standard | Mid layer | $0.939 \pm 0.01878$ | $0.883 \pm 0.05348$ |
| Metamer | Standard | Late layer | $0.310 \pm 0.02817$ | $0.121 \pm 0.01388$ |
| Metamer | L2 | Pixel space | $0.998 \pm 0.00122$ | $0.989 \pm 0.00653$ |
| Metamer | L2 | Mid layer | $0.997 \pm 0.00163$ | $0.985 \pm 0.00816$ |
| Metamer | L2 | Late layer | $0.994 \pm 0.00245$ | $0.990 \pm 0.00694$ |
| SnS | Standard | Pixel space | $0.397 \pm 0.087$ | $0.224 \pm 0.014$ |
| SnS | Standard | Mid layer | $0.476 \pm 0.099$ | $0.256 \pm 0.022$ |
| SnS | Standard | Late layer | $0.779 \pm 0.070$ | $0.636 \pm 0.024$ |
| SnS | L2 | Pixel space | $0.843 \pm 0.052$ | $0.936 \pm 0.040$ |
| SnS | L2 | Mid layer | $0.722 \pm 0.035$ | $0.857 \pm 0.041$ |
| SnS | L2 | Late layer | $0.721 \pm 0.030$ | $0.888 \pm 0.009$ |

Table S4: **Comparison of average distances with the reference natural image between metamers and SnS invariances**. Mean $\pm$ SEM are reported for each representation layer used in the experiments (`Pixel space`, `Mid layer`, and `Late layer` of ResNet50 standard and $L_2$-robust variants). Distances between natural images belonging to the same ImageNet category are reported as reference. For every network layer, SnS invariances yield substantially larger distances than both the natural-image baseline and the metamers.

| Condition | Net Layer | ResNet50 | ResNet50 ($L_2$) |
|---|---|---|---|
| Metamer | Pixel space | $0.7998 \pm 0.01523$ | $0.7998 \pm 0.01523$ |
| Metamer | Mid layer | $978.2 \pm 6.483$ | $6.181 \pm 0.5169$ |
| Metamer | Late layer | $356.8 \pm 10.48$ | $8.254 \pm 0.9402$ |
| SnS | Pixel space | $281.4 \pm 5.991$ | $281.8 \pm 4.94$ |
| SnS | Mid layer | $1935 \pm 8.864$ | $371.8 \pm 2.603$ |
| SnS | Late layer | $858.2 \pm 22.75$ | $406.3 \pm 6.442$ |
| Natural | Pixel space | $143.4 \pm 1.447$ | $142.4 \pm 1.669$ |
| Natural | Mid layer | $1478 \pm 8.138$ | $201.3 \pm 2.268$ |
| Natural | Late layer | $384.8 \pm 7.128$ | $239.1 \pm 1.845$ |

