# OpenReview forum: "Stretching Beyond the Obvious: A Gradient-Free Framework to Unveil the Hidden Landscape of Visual Invariance"
_ICLR.cc/2026/Conference — ICLR 2026 Poster_

### Official Review · Reviewer_JP16 · 2025-11-01

**Soundness:** 3
**Presentation:** 3
**Contribution:** 4
**Rating:** 6
**Confidence:** 5

**Summary:**

This paper proposes a new method—Stretch and Squeeze (SnS)—for testing the invariance and adversarial sensitivity of neurons within deep neural networks. The approach can be applied to any two layers of a network to examine both upstream invariance and downstream adversarial sensitivity. The authors demonstrate that the SnS-generated invariant and adversarial images outperform those generated by simple affine transformations, and that these images better align with human perception. Applying this method also reveals differences between robust and standard models. The framework has potential applications for understanding the tuning properties of biological neurons.

**Strengths:**

1. The study unifies the measurement of neural invariance and adversarial sensitivity into a single computational framework, which is conceptually novel.
2. The proposed method could serve as a useful tool for probing biological neural properties in future neuroscience research.
3. The writing is clear and the structure is easy to follow.

**Weaknesses:**

1. Although the paper is categorized under “Application to Cognitive & Neuroscience,” it does not actually include experiments or validations on biological neural data.
2. The paper identifies differences between robust and standard networks but does not attempt to explain the origin or meaning of these differences.
3. The figure quality can be substantially improved.

**Questions:**

1. In the abstract, the authors claim that “adversarial training fails to increase the interpretability of high-level invariances”, but it is unclear how this conclusion was derived. It is just because its alignment with human perception is relatively low. This is interesting but why is the case? I understand the empirical observations, but I do not fully see how the presented results logically imply that adversarial training is ineffective. For example, in Figure 4C and 4E, differences between robust and standard models are apparent, but it remains unclear why these differences demonstrate the failure of adversarial training rather than simply reflecting distinct representational properties.

2. The authors show that SnS can produce invariant and adversarial images that outperform those generated by affine transformations. However, an affine transformation may not be the most informative baseline. Since SnS involves two complementary losses (stretch and squeeze), it would be more convincing to compare it against a single-loss variant, such as using only the stretch term to generate invariant images. This comparison could more clearly highlight the contribution and necessity of the full SnS framework.

---

> ### Author Response · Authors · 2025-11-24
> **Comments on the Weaknesses**
>
> We thank the reviewer for acknowledging the impact and significance of our work and for raising several key comments. We have addressed these issues in our rebuttal below and in the revised version of the manuscript that we have uploaded. Please note that changes from the initial submission pertaining to this review are highlighted in violet, with changes responding to other reviewers’ comments highlighted in different colors. The mathematical formulation of SnS in Section 3.1 has also been improved and clarified, but is not highlighted for readability.
>
> ## 1. Validation on biological neural data
> We agree with the reviewer that validating SnS with biological neural recordings is an exciting direction, and it is one that our present work is designed to directly enable. The primary contribution of this paper is to establish and validate SnS as a novel framework for probing the invariance and robustness of neural units. This foundational work is a necessary prerequisite to motivate and justify its application to future experiments with non-human primates, rodents or human participants. To this end, we have developed a method that extends the gradient-free optimization style introduced by XDream (Xiao et al PLoS Comp Bio 2020, Ponce et al Cell 2019) - a framework to discover Most Exciting Images (MEIs) that has been successfully validated in CNNs and applied to macaque visual cortical neurons. In addition, we provide three key biologically-relevant forms of validation of SnS:
>
> a. *Computational validation in demonstrably “brain-like” models.* We applied SnS to a range of architectures (e.g., ResNet, VGG and ViTs) that are among the leading image-computable models of the primate visual system according to benchmarks like Brain-Score (Schrimpf et al., BioRxiv 2018).
>
> b. *Validation on target units in hidden layers.* We applied SnS to uncover the invariant images of ResNet50 units in intermediate convolutional layers (see Section E in the Supplementary Material and supplementary Fig. S6). The tuning of these units is relatively consistent with that of neurons in cortical visual areas. In fact, biological neurons (even in the inferotemporal cortex, the highest visual cortical area in primates) have receptive fields (RFs) with a limited spatial extent and do not encode such abstract semantic images as those encoded by readout units of CNNs trained with Imagenet categories. Interestingly, our tests show that SnS not only discovers the invariance fields of these units but also isolates the specific portions of the image plane that these units process (Fig. S6) - i.e., the equivalent of the RFs for biological neurons. In this sense, SnS is superior to XDREAM, which yields MEIs that span the entire image plane, even when the target units actually process only small subregions of it (Fig. S6). This observation suggests that SnS could be a very powerful tool to explore the tuning and invariance fields of biological visual neurons.
>
> c. *Showing that hierarchical invariances are robust to representation subsampling.* To assess SnS’s applicability to neuroscience, we generated the invariant images for a target downstream unit by stretching the representation provided by a heavily subsampled upstream layer (see Supplementary Material, Section F). This closely simulates a typical scenario faced by neurophysiologists, who are never able to record from *all* the neurons in a given visual area. Both qualitative inspection of the resulting invariant images and quantitative measurement of their representational distance from their MEIs revealed that the resulting invariances closely resembled those obtained using the full layer representation (Fig. S7). This strongly supports SnS’s potential for applications in neurophysiology.
>
> Overall, these experiments provide a series of clear, testable hypotheses about neural invariances that can now be directly investigated in future in-vivo experiments, thus relevant to “applications to neuroscience & cognitive science” category.
>
> ## 2. Discussion of the differences between robust and standard networks
> For a discussion of this issue, we refer the reviewer to our reply to their Question #1 below.
>
> ## 3. Figure quality
> We thank the reviewer for this suggestion. A​ problem we acknowledge but that is difficult to address is the small size of some panels in Fig. 4. This is due to the need of providing a complete comparison of the interpretability of the invariant images obtained for 3 different generator networks and classified by humans and 6 different observer networks, each coming in two flavours (robust and standard). In the supplementary materials, we have provided additional figures, with larger size and more example MEIs and invariant images (e.g., see the new supp. figure S4b). We will improve the figures in the camera-ready version, per the reviewer’s suggestion. Additionally, we would appreciate it if the reviewer can point us to some specific figure that was particularly unclear.

---

> ### Author Response · Authors · 2025-11-24
> **Reply to the Questions (1/2)**
>
> ## 1) Derivation of the claim that “adversarial training fails to increase the interpretability of high-level invariances”
>
> We thank the reviewer for pointing out that this conclusion of our work was not explained clearly enough. Our conclusion that “adversarial training fails to increase the interpretability of high-level invariances” stems from two observations reported in Fig. 4.
>
> 1) *Alignment with human perception was excellent when examining the MEIs generated for the robust ResNet50.* Human participants classified these MEIs with accuracy comparable to that achieved on the natural images encoded by the corresponding readout units (compare the cyan box to the blue box in Fig. 4b). By comparison, human classification of the MEIs obtained for the standard ResNet50 was very poor (pink box, Fig. 4b). Therefore, considering only the interpretability of the MEIs, one would conclude that the robustification process was very effective in increasing the alignment of image classification between ResNet50 and human observers.
> However, what SnS shows is that this alignment gradually deteriorates when considering invariant images obtained by stretching the representation in progressively deeper layers (green boxes, Fig. 4b). At the same time, paradoxically, the interpretability of higher-level invariances slightly improves for the standard network (red boxes, Fig. 4b). As a result, human interpretability of the invariant images obtained by stretching high-level representations is only marginally better in the case of the robust network compared to the standard one. This finding supports our conclusion that “adversarial training fails to increase the interpretability of high-level invariances”.
>
> 2) Further support to this conclusion comes from the same kind of interpretability tests, but performed on other observer networks (Fig. 4c-e). When these observer networks are themselves robust (left panels in Fig. 4c and e), the resulting trends are very consistent with the one found in humans - hence the very large correlations between humans and networks shown in Fig. 4d (dark bars). This demonstrates that, despite the robustification process, the tuning/invariance landscape of ResNet50 readout units remains idiosyncratic and not fully generalizable to other human and artificial observers. This observation holds true also when standard networks are used as observers, although, in this case, the agreement between human participations and networks is, in general, lower (see Fig. 4d, light bars).
>
> We agree that, in our original manuscript, the description of these results was too concise. **In our revised version, we have expanded it**, trying to make it clearer why our experiments support the conclusion of a failure of adversarial training to increase the interpretability of high-level invariances **(see our revised Section 4.3)**.

---

> ### Author Response · Authors · 2025-11-24
> **Reply to the Questions (2/2)**
>
> ## 2)  Comparing SnS with single-loss variants
> We thank the reviewer for this suggestion. We agree that comparing SnS to variants that use only one loss term at a time provides an informative ablation experiment and helps clarify the necessity of the dual-loss formulation of SnS. To address this, **we repeated the experiment shown in Fig. 2** (i.e., targeting a readout unit and using pixel space as the representation space) with four single-loss optimization variants: squeeze-up (minimize the distance in target-unit activation relative to the reference MEI), stretch-up (maximize that activation distance), squeeze-down (match the pixel-space representation of the MEI), and stretch-down (maximize pixel-space distance from the MEI). As in Fig. 2, we evaluated 77 target units with 10 random seeds each, yielding 770 runs per condition. Squeeze-up and stretch-down followed the initialization and Pareto-ordering protocol used for SnS invariances; squeeze-down and stretch-up followed the procedures of SnS adversarial experiments (for a detailed description see Section 3.1 and Supplementary material, section A.3).
>
> Overall, none of the single-loss variants achieved the characteristic trade-off that SnS obtains for invariant or adversarial image synthesis. Optimizing only one term consistently failed to drive the other metric into the appropriate regime, underscoring the functional necessity of the dual-loss structure. Nevertheless, each single-loss optimization produced a characteristic behavioral pattern worth highlighting **(see the supplementary Fig. S5 in Section D of the Supplementary Material)**:
>
> - *Squeeze-up.* This optimization increases target-unit activation into the invariant regime (i.e., near the MEI level), but pixel-space distance from the MEI remains close to that of the initial random images with which the optimization is initalized (see the black curve in the **new Fig S1b**, right; iteration = 0). This behavior closely resembles running XDREAM with multiple random initializations: it yields multiple images that strongly activate the unit while differing from one another in pixel space, but does not discover maximally invariant solutions that explore the edge of the invariance manifold, like SnS does.
>
>
> - *Stretch-up.* Maximizing deviation in unit activation naturally drives divergence in pixel-space distance. In other words, suppressing the target unit requires making the image more dissimilar from the MEI. This behavior is essentially the opposite of SnS adversarial synthesis, which achieves strong activation suppression with minimal pixel-space change.
>
>
> - *Squeeze-down.* Matching the MEI in pixel space also preserves high target-unit activation. This objective is closely related to the generation of  model metamers (Feather et al., Nature Neuroscience 2023), which match internal representations at intermediate layers of the CNN. In this sense, metamer synthesis can be viewed as a special case that fits naturally within the SnS dual-loss framework.
>
>
> - *Stretch-down.* Stretching only the pixel-space representation causes the image to diverge from the reference leaving the activation of the target unit close to its initial value at the beginning of the optimization (see the black curve in **Fig. S1b**, left; iteration = 0), which is very far from the one produced by the MEI.
>
> **We have added these results to the revised manuscript in the new Supplementary Section D, mentioning them in the result section 4.1.**

---

> ### Comment · Reviewer_JP16 · 2025-11-27
>
> I thank the authors' responses. It sounds OK to me. I decide to keep my score.

---

### Official Review · Reviewer_ac3Z · 2025-11-01

**Soundness:** 3
**Presentation:** 3
**Contribution:** 3
**Rating:** 6
**Confidence:** 5

**Summary:**

This paper introduces Stretch-and-Squeeze (SnS), a novel, gradient-free, and model-agnostic framework for systematically characterizing the invariance landscape and adversarial sensitivity of units in both artificial and biological visual systems. SnS frames these as bi-objective optimization problems using evolutionary algorithms (specifically CMA-ES) operating on the latent space of a generative model.

For probing invariance ($\Xi_{inv}$), SnS seeks to maximize the distance (stretch, $L_{stretch}^{\kappa}$) between a candidate image and a reference image in a chosen intermediate representation space ($\kappa$), while simultaneously minimizing the change (squeeze, $L_{squeeze}^{l}$) in the activation of a target unit downstream ($l$). The dual objective is used for generating adversarial examples ($\Xi_{adv}$).


Applying SnS to standard and adversarially trained CNNs (ResNet50), the authors make several significant discoveries:

1. Hierarchical Invariance: The nature of the discovered invariant images varied dramatically depending on the representation layer being stretched: pixel-level changes mainly affected luminance/contrast, mid-layers altered texture, and late-layers altered pose.

2. Interpretability of Robust vs. Standard Models: Adversarially trained (robust) networks yielded invariant images that were initially more human-interpretable than those from standard models. However, this advantage eroded or reversed at deeper layers: robust network invariances became less interpretable when stretching deeper, while standard network invariances became more interpretable.

3. Conclusion on Robustness: This suggests that adversarial training, while improving pixel-level perceptual alignment, fails to increase the interpretability of high-level invariances.

The framework's versatility is demonstrated by its application to different CNN architectures (ResNet18, VGG16_bn) and Vision Transformers (ViT), and its potential utility in visual neuroscience experiments by showing efficacy even when only a small fraction of units are recorded.

**Strengths:**

* Originality and Technical Novelty: SnS is a gradient-free, model-agnostic framework for systematically exploring the full invariance manifold of a unit, moving beyond local measures or pre-defined transformations. The bi-objective optimization scheme is elegant and powerful, unifying the search for invariant images and adversarial examples.

* High Quality and Rigor: The experimental design is robust. The authors not only apply the method to a benchmark (ResNet50) but also validate the findings across different architectures (ResNet18, VGG16_bn, ViT) and use a multi-observer and human classification setup for interpretability.

* Clarity of Insight: The results are not just empirical observations but yield a deep conceptual insight: the stark, hierarchical divergence in interpretability between standard and robust networks at deeper layers. This challenges existing conclusions about adversarial robustness and perceptual alignment.

* Broad Significance: The technique is directly applicable to visual neuroscience, offering a method to probe biological neurons without relying on perfect "digital twin" fidelity, which is a major constraint in current gradient-based neuro-visual studies.

**Weaknesses:**

* Computational Cost: The reliance on evolutionary algorithms (CMA-ES) is a known trade-off for gradient-free and model-agnostic optimization. The computational cost can be very high, especially given the search space dimension ($n=4096$). While the results are excellent, the practical utility of SnS for very large-scale or high-throughput experiments may be limited compared to gradient-based methods.

* Generative Model Dependency: The quality and expressivity of the invariant images are fundamentally constrained by the generative model ($\psi$) used. The authors assume a "powerful prior" is embodied in the model. The choice of generative model, and its potential bias on the invariant manifold, warrants further discussion or empirical analysis.

* Convergence Analysis: The paper notes that a significant percentage of runs did not reach the maximum iteration count (up to 86.75% for some standard network conditions). While the authors state the final populations still reached "functionally relevant activation regimes", a more detailed discussion on the impact of non-convergence or early stopping on the resulting Pareto front quality would strengthen the completeness of the method's presentation.

* References: The authors also missed citing some critical works in this domain. For e.g. Extreme Image Transforms (EITs) [Crowder et al., 2022; Malik et al., 2023, Biol Cybernetics, Malik et al., 2023, arXiv] which present a novel view of structural changes in the input images, and MIRC [Ullman et al., 2016, PNAS] which present similar experiments with human and network observers. Authors should also mention a comparison of their method to that of EITs given they also make changes to the image space at varying levels and test against VOneResNet [Dapello et al., 2020, NeurIPS] that claims to explain the V1 variance, showing that EITs outperform VOneResNet by a significant margin.

**Questions:**

1. Can the authors elaborate on the choice of the generative model $\psi$? Specifically, how might a different generative model (e.g., a modern diffusion model vs. the cited Dosovitskiy & Brox, 2016 model) affect the sampled invariant manifold, and have the authors explored this?

2. The paper focuses on an $L_2$ robust ResNet-50 ($\epsilon=3$). Would the striking hierarchical divergence in interpretability also hold for $L_\infty$-robust models, which tend to learn different features?

3. Given the bi-objective nature, the results rely on the final solution from the Pareto front. Could the authors provide a more detailed analysis or visualization of the entire Pareto front for a few example units to demonstrate the trade-off space between stretch and squeeze?

4. The method is described as model-agnostic. Have the authors considered or experimented with non-vision applications, for instance, characterizing invariances in audio or text models?

5. Authors should mention the IRB number at the end of the main text and consider releasing code on a public platform. Authors should also provide details on the training and hardware setup.

6. Have the authors tested this approach with larger networks that ResNet and ViTs? The authors should write a remark if they cannot make a direct comparison.

7. Have the authors considered evaluating the human alignment with any fMRI or EEG datasets (like Natural Scenes Dataset (NSD)), especially in case of the digital twin analogy?

8. I understand the author information was redacted for anonymity from line 492 in the reproducibility statement. Just a reminder to fill it in for the final version.

---

> ### Comment · Reviewer_ac3Z · 2025-11-20
>
> Please let me know if there are any questions about the review. Thanks.

---

> > ### Author Response · Authors · 2025-11-21
> >
> > We thank the reviewer for the message. We are currently finishing our  rebuttal and will post it by the end of today Anywhere on Earth (A.O.E)

---

> ### Author Response · Authors · 2025-11-22
>
> We thank the reviewer for acknowledging the impact and significance of our work and for raising several key comments. We have addressed these issues in our rebuttal below and in the revised version of the manuscript that we have uploaded. Please note that changes from the initial submission pertaining to this review are highlighted in blue, with changes responding to other reviewers’ comments highlighted in different colors. The mathematical formulation of SnS in Section 3.1 has also been improved and clarified (following concerns of Rev N9Ge), but is not highlighted for readability.

---

> ### Author Response · Authors · 2025-11-22
> **Comments on the Weaknesses**
>
> ## Computational cost
>
> We agree with this concern of the reviewer. The computational cost can be a limiting factor in the application of our method, especially in neurophysiology experiments, where the activity of large neuronal populations can typically be recorded only for a limited stretch of time. On the other hand, it is exactly in these scenarios that gradient-free methods such as SnS are necessary, since they bypass the need for a  faithful digital twin typical of gradient-based approaches (see Section 2).
>
> Returning to the issue of the computational cost of this approach, when using high-density electrode arrays in high-throughput neuronal recording experiments, one way to mitigate it would be to apply SnS to multiple target units at a time, thus parallelizing the search for adversarial/invariant images within the same optimization loop. This should be achievable thanks to the fact that each neuron has its own receptive field (RFs). If the RFs of the optimized neurons are non-overlapping, this would allow SnS to run independently on different portions of the presented images, discovering multiple invariant/adversarial examples in one optimization run.
>
> Conversely, in case of neurons with largely overlapping RFs (i.e., units processing the same portion of the input images), one could first screen the recorded units using natural images (e.g., as done with XDream by Ponce et al, 2019) to search for subpopulations of neurons with similar selectivity. SnS could then be applied to such subpopulation to find the collective invariance field of the neuronal ensemble. **In our revised manuscript, we have mentioned these possible further developments in the “Limitations and broader impact” subsection of the Discussion**.
>
> ## Generative model dependency
> For a discussion of this issue, we refer the reviewer to our reply to their Question #1 below.
>
> ## Convergence analysis
> We agree with the reviewer that this is indeed an important point that needs further discussion in our manuscript. As the reviewer correctly points out, our early stopping criterion (i.e. 90% of the evolved images with activations above the maximally activating natural image in ImageNet for invariance experiments and below the minimally activating image for adversarial ones) was developed to ensure that the activity of the target unit reached functionally relevant regimes. This heuristic criterion is particularly valuable because SnS is designed to operate on black-box systems (like biological brains), where theoretical guarantees of convergence are not available, making such data-driven benchmarks essential.
>
> However, we want to underline that this criterion represents a very ambitious target, far exceeding what is needed for an image to be considered invariant (or adversarial). We intentionally chose such a strict threshold in order to rigorously stress-test the limits of our optimization approach, and to ensure that the resulting images were robust examples of invariance and not merely borderline cases. For example, let us consider a readout neuron tuned to the class  'goldfish': a reasonable activation level for an image to be considered invariant should be one that is comparable to or exceeds (like MEIs do) the distribution of responses to natural 'goldfish' images, and very distinct from the distribution of responses to non-'goldfish' images. Our early stopping criterion requires strongly exceeding the activations in this distribution. To address the concern of the reviewer regarding trials that did not converge to the early stopping criterion, **we have now added a new table (Table S1, Supplementary material, section A.5.1)** where we report the activation statistics of all the optimization runs that did not converge to the criterion. This table confirms that, even in such non-converging optimization runs, the target activations we achieve are clearly compatible with this conceptualization of invariance. Indeed, the activity of these synthesized invariant images on average vastly exceed those produced by typical natural images of the category to which the readout unit of interest is selective to (compare table data with the first set of boxes in Fig. S1a). Conversely, for adversarial attacks, the table shows that non-converging runs succeed at silencing the unit completely (i.e., bringing the activity below 0). Moreover, **in the new supplementary Fig. S0** (i.e., examples of full Pareto fronts for invariant and adversarial experiments) we show, for each task (i.e. invariance /adversarial attack) and network (robust and standard Resnet50), an example of convergence to the criterion and an example front of a run that did not converge to the criterion.
>
> ## References
>
> We thank the reviewer for pointing out these relevant references. We are currently studying the papers and we will certainly add a comparison with SnS in a further revision of our manuscript that we will upload by the beginning of the next week.

---

> ### Author Response · Authors · 2025-11-22
> **Reply to the Questions (1/2)**
>
> ## 1. Choice of the generative model.
> We agree with the reviewer that, given the modularity of our framework, one could in principle use any generator, including modern large-scale GANs or diffusion models.
>
> The characteristics of the generative model are an important determinant of results derived from SnS, since the generative model effectively constrains the image-space manifold where invariances and adversarial examples are found. Ideally, one would like this manifold to be that of natural image statistics, but without biasing the generator to  “hallucinate” exceedingly complex photorealistic details which are unlikely to be encoded by individual visual neurons, especially in low- and mid-level processing stages.
>
> With this specific goal in mind, our choice of the generative model from Dosovitskiy & Brox (2016) was a deliberate decision guided by two related factors: consistency with previous related work (XDream), and the specific nature of that generator's prior.
>
> First, our work extends the gradient-free optimization style introduced by XDream to new scientific goals (the joint search for adversarial and invariant stimuli). XDream has been successfully applied to both artificial and biological neural systems (Ponce et al., 2019; Xiao & Kreiman, 2020), and these studies were specifically validated using the Dosovitskiy & Brox generator. Retaining this generator allowed us to build directly on an established methodological foundation.
>
> Second, and crucially, this generator embodies the balance we require: it enforces a naturalistic prior that is not overly restrictive. While modern large-scale GANs and diffusion models excel at photorealistic generation, they achieve this by enforcing extremely strong priors that can be overly confining for our goals. This concern is supported by empirical evidence gathered by Wang and Ponce (Wang and Ponce, bioRxiv 2024), who directly compared XDream optimization using BigGAN versus Dosovitskiy & Brox while performing neuronal recordings from monkey visual cortex. They reported that “BigGAN was considerably less abstract due to its bias towards creating sharply focused objects, even for V1” (i.e., a low-level visual cortical area). Given this evidence, we chose to stick with a generative model with a weaker prior. This ensures that the generated images are shaped primarily by our scientific objective, rather than being confined by a pre-learned and potentially restrictive image manifold. However, despite this difference, Wang and Ponce, 2024 show that, in many other cases, the overall activation performance and many representational trends were similar across the two generators, suggesting that different generators are often interchangeable. We think that this kind of comparison (extended also to diffusion models) would be an interesting direction for future work, but it would require a comprehensive comparative study that falls outside the  scope of the current manuscript. **In our revised manuscript,  we added a short paragraph with these considerations in the Discussion section.**
>
> ## 2. Does the hierarchical divergence in interpretability hold for L_inf-robust models?
> This is a very interesting question. We are currently running new experiments using an L_inf robust ResNet-50 as the generator network. We expect to complete these tests in the next few days and be able to reply to the reviewer’s point by the beginning of the next week, when we will also upload a further revision of our manuscript.
>
> ## 3. Visualization of the entire Pareto front for few example units.
> We thank the reviewer for raising this point. **We have added a dedicated visualization of the final Pareto fronts for several representative units in the new supplementary figure Fig. S0.**

---

> ### Author Response · Authors · 2025-11-22
> **Reply to the Questions (2/2)**
>
> ## 4. SnS: non-vision applications (e.g. audio or text).
> We thank the reviewer for this excellent suggestion. We agree that, given that our method is model-agnostic, it is potentially applicable to uncover invariances in other domains (i.e., other sensory modalities). This consideration applies equally to XDREAM, the framework that SnS stems from, which, to the best of our knowledge, has so far only been used in the visual domain.
>
> Conceptually, both SnS and XDREAM could be extended to modalities such as audition. For instance, one could characterize the preferred stimuli of auditory neurons (with XDREAM) or identify their invariant and adversarial stimuli (with SnS). The key requirement for such an extension would be the availability of an effective generator capable of producing audio waveforms or spectrograms from a low-dimensional latent code. Suitable candidates do exist (e.g. SpecGAN (Donahue et al., arXiv 2018), WaveGAN (Donahue et al., arXiv 2018)). With respect to the optimizer, latent spaces would integrate naturally with our CMA-ES optimizer, which operates directly in the generator’s code space.
>
> Running these kinds of experiments certainly could yield fascinating results. However, experimental validation in non-visual domains would require setting up entirely new generative pipelines and baselines. We believe this would be a distinct research undertaking better suited for future work, as we aim here to provide a comprehensive analysis within the visual domain. Accordingly, **we have mentioned these potential future developments in our revised Discussion.**
>
> ## 5. IRB number, code release, training and hardware setup.
> We agree that it is important to disclose IRB information and to publicly release the project’s source code. We refrained from this in the initial submission to maintain anonymity, but will add the IRB number and public code repository in the camera-ready version. **We provide details on our model training and hardware setup in Supplementary Section A.6**. If the reviewer is referring to training and hardware with regards to human participants, **we provide details on how participants were introduced to the task in the 3rd paragraph of Supplementary Section B** (“Human Subjects Experiment”; there was an initial warmup and screening phase for each participant before data collection began in earnest). **We also updated the same section** to further specify that the online (Prolific) participants were required to complete the tasks on a desktop or laptop computer, with access via smartphones and tablets blocked programmatically to ensure consistent stimulus presentation.
>
> ## 6. SnS with larger networks.
> This is a great suggestion for validating the generalizability of SnS. Our work demonstrates SnS’ generative capabilities using eight different neural models: VGG16_bn, ResNet-18, ResNet-50 (these 3 models were tested in both their standard and L2 robust versions), ImageNet-pretrained ViT-B-16, and CLIP-pretrained ViT-B-16. This validates SnS across three substantially different architectural families, parameter counts spanning roughly an order of magnitude (from 11.7M for ResNet-18 to 86.6M for ViT), and three distinct pretraining objectives (standard classification, adversarial training, and contrastive learning). We believe this validation across diverse model types provides a strong test of our framework’s generalizability.
>
> While we have not tested SnS on even larger models (e.g., DINO, ViT22-B) due to their substantial computation requirements, SnS is model-agnostic and relies only on forward passes. Since its mechanism is not dependent on model scale, we expect it to be directly applicable. **In our revised manuscript, we have added a remark to this effect in the “Limitations and broader impacts section” of the Discussion**.
>
> ## 7. Human alignment with fMRI and EEG Datasets.
> It is an excellent suggestion to consider validating our approach against large-scale neural datasets such as NSD. Since our SnS approach is designed to generate novel stimuli (i.e., invariant images), evaluating it against a static, pre-existing neural dataset such as NSD would require training and validating a dedicated neural encoding model using the dataset and then evaluating its responses to our generated images. While this would be a fascinating and powerful approach, this endeavor is beyond the scope of the present work. Notably, we do evaluate the interpretability of invariant images using architectures (e.g., DenseNet, ResNet, VGG; see Figure 4) whose activations are known to strongly predict visual neural activity through benchmarking efforts such as Brain-Score (Schrimpf et al, BioRxiv 2018).
>
> ## 8. I understand the author information was redacted for anonymity from line 492 in the reproducibility statement. Just a reminder to fill it in for the final version.
> We appreciate this reminder, and will be sure to add the full author information on line 492 in the camera-ready version.

---

> > ### Comment · Reviewer_ac3Z · 2025-11-27
> >
> > Thank you for the detailed rebuttal response. i look forward to seeing these in the main text/appendix of the paper for further understanding of the work to the end user. i highly recommend comparing the methods (or justification not to) with the other papers mentioned above [EITs, MIRCs, etc.].

---

> > > ### Author Response · Authors · 2025-12-01
> > > **Last two pending questions (1/2)**
> > >
> > > We thank the reviewer for their engagement in our rebuttal. We provide below a reply to their last two pending questions. We have also uploaded a full revision of our manuscript addressing all of the reviewer’s comments (changes pertaining to this review highlighted in blue, changes responding to other reviewers’ comments highlighted in different colors).
> > >
> > > ## References
> > >
> > > We thank the reviewer for pointing out the relevance of the Extreme ImageTransforms (EITs) and Minimal Recognizable Configurations (MIRCs) literature. Both approaches emphasize the importance of studying machine vision through controlled image transformations grounded in principles from human vision. They highlight a critical limitation of current deep learning vision systems: these models often rely on features and recognition strategies that differ markedly from those used by humans. In this sense, they pursue goals that are indeed similar to those of SnS.
> > >
> > > However, SnS proceeds from the opposite direction. Rather than applying predefined transformations, we construct maximally invariant stimuli through generative stretching in latent space, while preserving class activations, and then test the interpretability of these invariances in human and ANN observers. Therefore, **unlike EITs, SnS does not prespecify the kind of transformations being applied to the images. Instead, the model’s own representational geometry determines the transformations SnS finds.** Although the methodology differs, we believe that placing SnS in the context of EITs and MIRCs suggests several promising future directions.
> > >
> > > First, it would be valuable to examine whether the human interpretability of SnS-derived invariances correlates with human alignment measures, derived by computing EITs or MIRCs. This is particularly relevant to the reviewer’s reference to VOneResNet. As shown in Malik et al. (2023), VOneResNet aligns poorly with human performance when probed with EITs. While we did not use VOneResNet as a generator network in our study, we did use several L2-robust CNNs, which are expected to be well aligned to human perception by the adversarial/robustification training. Our experiments show that, despite the robustification, the invariant images obtained for these networks are not always interpretable by human observers, particularly those obtained by stretching representations in deep layers. An intriguing open question is whether SnS would reveal a similar behavior for VOneResNet and how the two alignment metrics (SnS invariance interpretability vs. robustness to EITs) relate theoretically and empirically. We see these as compelling directions for future investigation into human–machine alignment across multiple metrics.
> > >
> > > Second, we agree that parametric image transformations represent an exciting future extension for SnS. In the present work, we focused on constructing invariances and adversarial examples through manipulations internal to the network’s own representational spaces (with the exception of pixel space). The resulting transformations, though systematically associated with the layer in which the stretching occurs (see Fig. 3), are complex and not easily expressible in simple parametric forms, unlike EITs. A natural next step would be to constrain SnS to operate within a parametric transformation space, such as that used in EITs. This would unify the interpretability and structural clarity of parametric approaches with the generative search capabilities of SnS, enabling the discovery of invariant or adversarial parametric stimuli tailored to a given network. From an efficiency standpoint, such a framework could substantially outperform grid search when exploring transformation parameters that preserve or disrupt perception.
> > >
> > > Given the limited space available, we were not able to discuss all these considerations in depth in our study, but **we did add a comment on EITs and MIRCs in Section 4.3 of our revised manuscript**, emphasizing their complementarity with SnS and suggesting a possible comparison between the two approaches.

---

> > > > ### Author Response · Authors · 2025-12-01
> > > > **Last two pending questions (2/2)**
> > > >
> > > > ## The paper focuses on an L2-robust ResNet-50. Would the striking hierarchical divergence in interpretability also hold for L-inf robust models, which tend to learn different features?
> > > >
> > > > We thank the reviewer for raising this interesting question. In this work, we focused our interpretability analyses on L2-robust networks, as L2 adversarial training is one of the most commonly studied and widely adopted forms of robustification. Across both human and model observers (six architectures, each in standard and L2-robust versions), we consistently observed a clear hierarchical divergence in L2 invariance interpretability, where high-level SnS invariances were substantially less interpretable than low-level ones. This pattern held across multiple generator architectures (ResNet-50, ResNet-18, VGG16_bn; Fig. 4e), supporting its generality under L2 robustification.
> > > >
> > > > As the reviewer correctly points out, L2 is just one of the  ways to robustify a network against adversarial attacks. Another popular method is L-inf robustification, which is done against L-inf bound adversarial attacks. The two kinds of adversarial training yield distinct outcomes: being robust to L-inf attacks does not guarantee robustness against L2 attacks and vice versa (Jiang et al., arXiv 2025).  To test how this different robustification method affects hierarchical invariances and their interpretability by other observer networks, we generated SnS invariant images for ResNet50 trained with L-inf robustification (epsilon = 4/255) obtained from RobustBench (Croce et al., 2020). Specifically, as in the experiment detailed in Fig. 4, we generated invariant images for n=120 natural images (the same as our original experiment) at three different layers of stretching (pixelspace, mid-level and high-level layers). Qualitatively (Fig. S10a) and quantitatively (Fig. S10b), layer-specific invariances in the L-inf-robust model remained highly distinct and discriminable in pixel space.
> > > >
> > > > To evaluate interpretability across observers, we presented the L-inf-robust invariances to three architectures (ResNet-18, Wide-ResNet-50-2, ConvNeXt-Base), each in both standard and L-inf robust versions (epsilon = 4/255, also from RobustBench). As in the L2 case, invariances from the L-inf robust generator network were on average more interpretable than those from a standard ResNet-50 generator, and interpretability again degraded when robust invariances were judged by standard networks (Fig. S10c). However, a key difference emerged: when L-inf invariances were evaluated by other L-inf robust models, classification accuracy either remained at ceiling or increased across stretching depths. This suggests that L-inf robust networks share highly aligned invariance structures, even at deeper layers.
> > > >
> > > > This interesting observation paves the way for future studies investigating and comparing invariances across different robustification regimes, but also of models that achieve “robustness” through explicit training to align with human perceptual strategies - e.g., harmonized neural networks (Fel et al., NeurIPS 2022) or, as the reviewer pointed out, VOneResNet, which tries to achieve robustness by incorporating a hidden layer matching the primate primary visual cortex (V1). As we emphasize in our comparison with the EIT and MIRC work, we believe that a combination of parametric approaches (e.g., EITs), gradient-based approaches (e.g., metamers), and gradient-free methods such as SnS will be essential for achieving a comprehensive understanding of invariances in deep neural networks - and how these differ across robustification strategies.
> > > >
> > > > **In our revised manuscript, we have described and discussed these new experiments towards the end of Section 4.3, as well as in the new supplementary Section J and the new supplementary Fig. S10.**

---

### Official Review · Reviewer_oZyT · 2025-11-03

**Soundness:** 3
**Presentation:** 3
**Contribution:** 2
**Rating:** 6
**Confidence:** 4

**Summary:**

The paper introduces Stretch-and-Squeeze (SnS), a gradient-free, model-agnostic framework to probe visual invariances and adversarial vulnerabilities in both DNNs and biological visual systems. SnS uses an evolutionary optimization strategy to get perturbations which stretch a representation, making it as different as possible, while squeezing a downstream activation, keeping it constant. Reversing these objectives will lead to adversarial examples. The authors claim an insightful finding with robust DNNs – invariant images are more human-recognizable at low levels but less so at deeper layers – which can be useful for further improving model adversarial robustness through human-model alignment.

**Strengths:**

- **Well-motivated framework, good practical value.** SnS focuses on understanding neural network representations, a fundamental challenge in the field. In comparison to prior works, SNS doesn’t look at unit activation but explore the space of transformations with well-grounded setting – a generative model, a test network, and a gradient-free optimizer. It is model-agnostic and gradient-free, particularly important with black-box models and neuroscience applications where gradient information is unavailable. It provides insights for model generalization and robustness.
- **Comprehensive experimental evaluation and clear presentation.** The paper includes experiments with various architectures such as ResNet-18, ResNet-50, VGG16, ViT, as well as human observers with rigorous statistical analysis. This shows the generalizability of SnS. Figures (e.g., figure 1 and 4) are well-presented to illustrate the key concepts and findings.
- **Interesting findings on adversarial training.** The divergent trends between robust and standard networks across layers (figure 4) is interesting. This hints that adversarial training does not uniformly improves human alignment and suggests that the robustness gain only align low-level, as opposed to high-level, invariances with human perception, which is informative for future investigation.

**Weaknesses:**

We thank the authors for submitting the paper to ICLR 2026! There are a few weaknesses listed below which I believe can make the paper better.
- **Lack of analysis on computational cost.** The paper uses the covariance matrix adaption evolutionary strategy for d(=4096)-dimentional codes but didn’t provide analysis of computational requirements, convergence properties, or comparisons with other gradient-based methods. How does this scale with network depth or code dimensionality? These considerations can be useful in practice.
- **Limited direct comparison to metamers or other invariance mapping techniques.** While the paper conceptually compares SnS with prior works such as metamers, empirical side-by-side comparisons would strengthen claims about its competence or unique coverage of the invariance space.
- **Lack of analysis on invariance manifold geometry.** While SnS generates multiple invariant images (figure 3), there’s limited analysis of the manifold structure itself. Information such as intrinsic dimensionality or whether the discovered invariances concentrated in certain directions or are uniformly distributed would be worth looking into.

**Questions:**

- How do you determine if optimization has converged? Do invariant images stabilize or do they continue to evolve? Some convergence curves would be good to see.
- For CNN experiments where gradients are available, how do SnS-discovered invariances compare to those found via gradient-based methods? This would help validate the gradient-free approach.
- Does the invariance found with different model architectures transferable? Are those more dependent on model architectures or training data or some other factors?

---

> ### Author Response · Authors · 2025-11-26
> **Comments on the Weaknesses (1/2)**
>
> We thank the reviewer for acknowledging the impact and significance of our work and for raising several key comments. We have addressed these issues in our rebuttal below and in the revised version of the manuscript that we have uploaded. Please note that changes from the initial submission pertaining to this review are highlighted in brown, with changes responding to other reviewers’ comments highlighted in different colors. The mathematical formulation of SnS in Section 3.1 has also been improved and clarified, but is not highlighted for readability.
>
> ## Comments on the Weaknesses
>
> ### 1)  Lack of analysis on computational cost
>
> We acknowledge that addressing the efficiency of SnS is important for broader adoption of the framework and we thank the reviewer for raising this point. In terms of computational complexity, both generator-based image synthesis from latent codes and the extraction of activations from the target network require only a single feedforward pass per iteration, whose computational cost, while depending on network size, is negligible in practice for all the architectures examined in our study, which span roughly an order of magnitude of parameter counts (from 11.7M for ResNet-18 to 86.6M for ViT). Consequently, the dominant computational bottleneck is the CMA-ES optimizer, whose running time scales quadratically with the dimensionality d of the search space. In practice, however, this does not pose a limitation: CMA-ES has been shown to be a highly effective method for gradient-free optimization of neural activity in vivo(Wang & Ponce, 2022a), and it remains computationally efficient under our hardware configuration as well. On our setup, a full optimization run (500 iterations) lasted approximately 120 seconds. As per the reviewer’s suggestion, we have added these considerations in the **revised version of the manuscript in Supplementary Material Sections A.1 and A.6**. Concerning the convergence of our optimization, we refer the reviewer to our reply to their Question #1 below.
>
> As a side remark, we would like to also point out that there are several possible ways to further improve SnS computational efficiency if needed. First, one could lower the maximum number of iterations or relax the early-stopping criteria (we used an extremely stringent stopping criterion for this study: please see also our response to this reviewer’s Question #1 below). Second, SnS is parallel across target units: runs for different units could be distributed over multiple GPUs or nodes, without changing the algorithm. Finally, SnS offers yet another way to reduce computational costs: robustness to representational space subsampling. As demonstrated in Supplementary Material Section F and Fig. S7, SnS recovers consistent invariance manifolds even when the representation space is heavily subsampled (down to 100 units). This suggests that, for large-scale experiments, the memory cost of storing the whole internal representation can be drastically reduced without loss of quality in the resulting images.
>
> ### 2) Limited direct comparison to metamers or other invariance mapping techniques.
>
> For a reply to this issue, we refer the reviewer to our reply to their Question #2 below.

---

> > ### Author Response · Authors · 2025-11-26
> > **Comments on the Weaknesses (2/2)**
> >
> > ### 3) Lack of analysis on invariance manifold geometry.
> >
> > We agree with the reviewer that analyzing the geometry of the invariance manifolds produced by SnS is an exciting research direction. Indeed SnS can be used to extensively sample the invariant space, paving the way for analyses on its intrinsic dimensionality and geometric structure.
> >
> > To demonstrate this potential, we added a new Supplementary Section H in which we perform an initial investigation of the geometry of SnS-derived invariance manifolds. We selected a single target readout neuron (the “chickadee’’ unit in the L2-robust ResNet-50 readout layer) and extensively sampled its maximally invariant images across three different stretching layers: pixel space (input/low-level), Layer3_conv1 (mid-level), and Layer4_conv7 (high-level). For each stretching layer, we ran SnS 100 times for each of 5 different reference MEI (obtained from independent XDREAM runs), yielding 500 invariant images per layer.
> >
> > To estimate the intrinsic dimensionality (ID) of each invariance manifold, we applied two distinct techniques to the vector representations of the invariant images: PCA, measuring the number of components explaining 95% of the variance; and two-nearest-neighbors intrinsic dimension estimation (2NN-ID; Facco et al., Scientific Reports 2017), a state-of-the-art nonlinear ID estimator that has been successfully applied to analyze image manifolds in both CNNs and visual cortical areas (Ansuini et al, Neurips, 2019; Muratore et al, Neurips, 2022). 2NN-ID was computed for different subsamples of data points (from n=50 to n=500 invariant images), repeating the process 50 times for each cardinality.
> >
> > The two approaches returned dimensionalities in very different numerical ranges: PCA required hundreds of components to explain 95% of variance (from 230 to 340 PCs; **see the new Fig. S9a**), whereas the ID estimated by 2NN-ID ranged between 10 and 30 across the three manifolds **(see the new Fig. S9b)**. This radical difference in ID estimation between a classic linear method (i.e. PCA) and 2NN-ID points to the fact that the structures of all 3 manifolds are highly nonlinear. This aligns with expectations for neurons high in the visual hierarchy, whose responses tolerate complex real-world transformations.
> >
> > Despite differences in scale, both methods agreed on a consistent ordering of dimensionality across stretching layers: lowest dimensionality for pixel-level stretching, highest for mid-level (Layer3_conv1) stretching, and reduced dimensionality again for high-level (Layer4_conv7) stretching.
> >
> > This pattern mirrors previously reported trends in the ID of image representations across deep CNNs (Ansuini et al., NeurIPS 2019), where the ID initially increases, peaks in mid layers and then gradually diminishes in deeper layers. This result further reinforces the distinction between invariances uncovered at different stretching layers (Fig. 3), indicating the ability of SnS to discover axes of variation that are distinct not only in qualitative terms (Fig. 3 A), but also in their dimensionality.
> > In our revised manuscript we mention this analysis in Section 4.2 and in the Discussion, while we illustrate it more extensively in the **new Section H of the Supplementary Materials and the new supplementary Fig. S9**.

---

> > > ### Author Response · Authors · 2025-11-26
> > > **Reply to Questions (1/2)**
> > >
> > > ### 1) How do you determine if optimization has converged? Do invariant images stabilize or do they continue to evolve? Some convergence curves would be good to see.
> > >
> > > The procedure to establish when the optimization converges is explained in Section A.5 of the Supplementary Materials. Our optimization could run for a maximum of 500 iterations, but we also implemented an early stopping criterion that worked as follows. In the case of searching for invariant images, the optimization would stop when 90% of the synthesized images produced activations above the maximally activating natural image in ImageNet for the targeted readout unit. Conversely, in the search for adversarial images, the optimization would stop when 90% of the synthetized images produced activations below the minimally activating natural image in ImageNet.
> > >
> > > This criterion represents a very ambitious target, far exceeding what is needed for an image to be considered invariant (or adversarial). We intentionally chose such a strict threshold in order to rigorously stress-test the limits of our optimization approach, and to ensure that the resulting images were robust examples of invariance and not merely borderline cases. For example, let us consider a readout neuron tuned to the class  'goldfish': a reasonable activation level for an image to be considered invariant should be one that is comparable to or exceeds - like Most Exciting Images (MEIs) do - the distribution of responses to natural 'goldfish' images, and very distinct from the distribution of responses to non-'goldfish' images. Our early stopping criterion requires strongly exceeding the activations in this distribution. As a result, in many instances, the early stop criterion was not reached and the optimization terminated at the maximum number of iterations (see the statistics reported at the beginning of Section A.5.1 in the Supplementary Material). Nevertheless, as shown in the supplementary Fig. S1a, in all our optimization runs the target activations we achieve with invariant images greatly exceed those produced by typical natural images of the category that the readout unit encodes (compare the last three sets of boxes to the first set). Similarly, the target activations we achieve with adversarial images are lower than those produced by typical natural images belonging to the categories that are not encoded by the readout unit (compare the third set of boxes to the second set).
> > >
> > > To make sure that this was the case also for those optimizations where the early stopping criterion was not reached, **we have now added a new table (Table S1, Supplementary material, section A.5.1)** where we report the activation statistics of all the optimization runs that did not converge to the criterion. This table confirms that, even in such non-converging optimization runs, the target activations we achieve are clearly compatible with this conceptualization of invariance. Indeed, the activity of these synthesized invariant images on average largely exceed those produced by typical natural images of the category to which the readout unit of interest is selective to (compare table data with the first set of boxes in Fig. S1a). Conversely, for adversarial attacks, the table shows that non-converging runs succeed at silencing the unit completely (i.e., bringing the activity below 0).
> > >
> > > Following the suggestion of the reviewer, **we have now added a new panel to the Supplementary Fig. S1 (i.e., Fig. S1b)** where we show the average convergence curves on both optimization goals (i.e., activation of the target readout units and Euclidean distance from the MEI in the pixel space) as a function of the iteration number. As shown in the figure, both goals converged to stable asymptotical values, which were consistent with the target optimization (i.e., search for invariant or adversarial images). It should be noted, in particular, that the curve showing the Euclidean distance from the MEI answers the reviewer’s question about whether the invariant images stabilized or continued to evolve. The Euclidean distance reached a stable value, thus showing that the images did not continue to evolve significantly.

---

> > > > ### Author Response · Authors · 2025-11-26
> > > > **Reply to Questions (2/2)**
> > > >
> > > > ### 2) For CNN experiments where gradients are available, how do SnS-discovered invariances compare to those found via gradient-based methods? This would help validate the gradient-free approach.
> > > >
> > > > This is also a very interesting question. To address it, we also took into account the previous comment of the reviewer concerning the limited direct comparison to metamers in our study. In fact, obtaining metameters is, arguably, the most powerful gradient-based approach to synthesize invariant images for CNN units that is currently available in the literature. We are currently running the code developed by Feather et al (2023) to produce metamers for Resnet50 (robust and standard) at network depths that are comparable to those used in our experiments with SnS. We plan to add a comparison between the invariant images produced by the two methods to the manuscript within the next few days or, in case working on this analysis takes too long, to the camera-ready version of the manuscript. If ready before the end of the discussion phase, we will post a comment here.
> > > >
> > > > ### 3) Does the invariance found with different model architectures transferable? Are those more dependent on model architectures or training data or some other factors?
> > > >
> > > > This is also an excellent question, and addressing it was actually one of the main goals of our study. A complete analysis concerning this issue is shown in Fig. 4, where, in addition to assessing how transferable to human perception the invariances discovered for Resnet50 readouts units are (Fig. 4a-b), we also measured how transferable those invariances are to other networks (Fig. 4c). The same tests were also performed for the invariances obtained for the readout units of two additional CNNs in Fig. 4e (Resnet18 and VGG16_bn) and for two vision transformers (ViT-B-16 with standard and CLIP training) in the supplementary Fig. S8 (see Section G of the Supplementary Materials). Overall, these experiments provide a complete comparison of the interpretability of the invariant images obtained for 5 different generator networks by humans and 6 different observer network architectures, each coming in two flavours (robust and standard).
> > > >
> > > > What these experiments show is that the invariant images obtained for a given generator network are only partially transferable to human perception and other observer networks and that the extent to which they are transferable crucially depend on three factors: 1) whether the generator CNN was robust or standard; 2) whether the invariant images were obtained by stretching the representation in initial (pixel-space input) or deep (nearer category output) layers; and 3) whether the generator network was a CNN or a vision transformer.
> > > >
> > > > In general, invariant images obtained for robust CNNs were more interpretable than those obtained for standard CNNs. However, the transferability of the invariances obtained for the robust networks gradually dropped when the representation was stretched in progressively deeper layers. By contrast, the interpretability of the invariances obtained for standard networks increased at high stretching depth. Vision transformers behaved more similarly to standard CNNs, but with the interpretability of the invariant images rising very sharply already when the stretching was applied in middle layers and remaining stable in deeper layers. As pointed out in our manuscript, this is consistent with the view that transformers learn less strictly hierarchical and more globally-integrated features than CNNs.
> > > > In summary, what these tests show is that the level of transferability of the invariances obtained for a generator network depends both on its architecture (i.e., CNN vs. vision transformers) and its training procedure (i.e., robustified vs. standard, in the case of CNNs). **To make this more explicit and to directly address the reviewer’s concern, we added a paragraph at the end of section 4.3 in our revised manuscript.**

---

> > > > > ### Author Response · Authors · 2025-12-01
> > > > > **Comparison to gradient-based method (metamers)**
> > > > >
> > > > > We provide below the reply to the last pending question raised by the reviewer.
> > > > >
> > > > > ***Limited direct comparison to metamers or other invariance mapping techniques.** While the paper conceptually compares SnS with prior works such as metamers, empirical side-by-side comparisons would strengthen claims about its competence or unique coverage of the invariance space. … For CNN experiments where gradients are available, how do SnS-discovered invariances compare to those found via gradient-based methods? This would help validate the gradient-free approach.*
> > > > >
> > > > > As anticipated to the reviewer, we carried out a comparison between SnS-derived invariant images and model metamers generated using the original implementation of Feather et al. (2023). Metamers were synthesized for the same n = 120 natural images used in our behavioral experiment (Fig. 4), ensuring that for each natural image we obtained one SnS invariant and one metamer at each of three depths (pixel space, mid-level, and high-level). All metamer-generation hyperparameters followed those reported by Feather et al. Metamers were generated using both a standard ResNet-50 and an L2-robust ResNet-50.
> > > > >
> > > > > We first replicated the interpretability experiment shown in Fig. 4c, but applied it to the metamer stimuli. Using the same observer networks as in our main experiments (six architectures, each in both standard and L2-robust versions; n = 12 total), we recovered results consistent with Feather et al. (Table S3). Specifically, metamers produced by the L2-robust ResNet-50 remained highly interpretable across observer models at all synthesis depths (pixel space, layer 3, layer 4). In contrast, standard-network metamers showed the expected sharp decline in interpretability for late-layer metamers (layer 4) across all observer networks. Critically, this pattern is the opposite of what we observe for SnS hierarchical invariances (Fig. 4c; numerical values in Table S3). Indeed, for standard networks, late-layer SnS invariances show a pronounced increase in interpretability, to the extent that observer accuracies far exceed those obtained for late-layer metamers from the standard network.
> > > > >
> > > > > This divergence in interpretability is accompanied by substantial differences in distance to the corresponding reference natural images (Table S4). Metamers are explicitly optimized to minimize representational distance to the target image at a given layer, whereas SnS invariances explicitly maximize that distance. As a result, metamers lie much closer to their reference images than SnS invariances at all depths and for both standard and robust generator networks. For comparison, we also report the average pairwise distances between natural ImageNet images from the same class at the same three layers. While metamers lie far below these natural-image baselines, SnS invariances systematically exceed them by a wide margin (Table S4).
> > > > >
> > > > > As detailed in the new Supplementary Section D, we further show that metamer-like stimuli arise as a special case of the SnS objective: applying a pure “squeeze’’ loss to a target representation at a given layer (i.e., the “squeeze-down’’ variant in Fig. S5) effectively reproduces metamer-like images. In this sense, SnS provides a gradient-free generalization of metamer synthesis, enabling analogous invariance characterization in settings - such as in vivo experiments - where gradients are not accessible.
> > > > >
> > > > > Finally, in terms of computational cost, generating 120 metamers required approximately the same time as generating 120 SnS invariants using the same hardware (Section A.6), indicating that the two algorithms are comparable in practical runtime.
> > > > >
> > > > > **In our revised manuscript, we have presented and discussed these new experiments in Section 5 (the Discussion), as well as in the new supplementary Section I and the new supplementary Tables S3 and S4.**

---

### Official Review · Reviewer_N9Ge · 2025-11-03

**Soundness:** 3
**Presentation:** 2
**Contribution:** 2
**Rating:** 2
**Confidence:** 4

**Summary:**

The authors present a framework for examining invariances and adversarial vulnerabilities of a CNN representation, in which
base images are modified simultaneously to maximize euclidean distance to the base in one layer while minimizing it in another.

They demonstrate through examples that the method can generate adversarial and invariant images, and that these are recognizable by humans or other networks.

**Strengths:**

The topic is important: the ML community needs tools for characterizing fundamental aspects of network function.

The general approach taken by the authors is well motivated and novel (although see below for some related literature that should be cited).

**Weaknesses:**

I like the conceptuatlization, but the construction of the method is ad hoc and not well explained, and the experimental results are somewhat anectdotal and not fully convincing.

First, the method.  I thought I understaood the basic construction after reading abstract/intro, but then found the more precise description in the  Methods section confusing.  Some specifics:

- Eq. (2) is a non-standard notation for a dual objective: Presumably one wants to either trade these two terms off (in which case they should be added, with a multiplier), or one wants to hold one fixed while minimizing the other.  Alternatively, one could take the quotient.  As written, I can't tell what was done for the examples in the papepr (see below).

- The methods section starts with description of a 3-component implementation, but provides no motivation for this. Specifically, why is there a need for a separate generative model?  Or a separate gradient-free optimizer?  I would think a unified algortihm, in which adjustments to the input are made by backpropagating gradients of the objective through the test network, would be easier to implenent, test, and interpret.

The empirical results:

- The method is stated in intro/abstract as seeking perturbations that trade off the L2 norm of activations at two different levels {k,l} of a network. But as far as I can tell, all experiments set one of these "levels" to be the input (level 0), and it's not clear to me how one would interpret a tradeoff bertween two internal levels.  I think readers would be better served by defining the method with respsect to images, and mentinoning the potential generalization in the Dicussion.

- The methods is defined in terms of the MSE of two layers, but most of the experiments seem to refer to individual readout units.  Why not MSE of the readout layer (or layer before)?

- Figure 2 shows a single example image (an MEI), with a  corresponding adversarial and invarient example.  Although it succeeds in demonstrating that the optimization is pushing the examples in the right direction (the ratios of pixel MSE and readout reduction are opposite), the images do not seem  perceptually convincing. Why was an MEI used for this (rather than a photograph)?  Also, I can't understand why the vertical axis is not Euclidean distance in readout layer (as used in the objective).

- Figure 3 shows examples of invariant images for a single MEI, computed for three different layers.  The text describes these as providing evidence that they corespond to luminance/contrast changes, texture/color changes, and viewpoint or multiplicity of object changes.
I don't find this very  convincing: The first row includes color changes, size chnages and multiplicity changes.  The third row inclues color chnages
Also, not clear why MEI's were used for the base images in this experiment.

- Since the combination of the two objective components is not specified, I could not understand the interpretation of the perceptual study. To make comparisons of performance on invariant images generated for different layers, it would seem essential to match the conserved objective (output layer?), rather than allow them to "float" along the pareto front (as described in end of sec 3.1).  Again, I could not find an explanation of how the two objectives were combined or controlled.

- Related work: the approach bears similarity to two methods that have been proposed for comparing models by searching for stimuli that best discriminate them (in some cases, holding one model fixed while maximizaing/minimizing the other): "controversial stimuli" (Golan et al 2020), and "MAD competition" (Wang et al 2008).  These should probably be cited in the "related works" section.

**Questions:**

See above.

---

> ### Author Response · Authors · 2025-11-20
> **Rebuttal to Methods issues**
>
> We thank the reviewer for acknowledging the novelty and timeliness of our work and for raising several key questions. Many of these questions concern clarifications about our method and the way it was applied in our tests. We have addressed these issues in our rebuttal below and in the revised version of the manuscript that we have uploaded. Please note that changes from the initial submission are highlighted in red. The mathematical formulation of SnS in Section 3.1 has also been revised, but is not highlighted in red for readability.
>
> ## Methods issues
> ### 1) Eq. (2) as non-standard notation for a dual objective
>
> We believe the reviewer's observations stem from a misunderstanding of the employed optimization process. Our SnS framework solves a bi-objective (vector-valued) optimization problem, which is conceptually distinct from the more standard scalar optimization problem where the loss function is a single scalar value, potentially resulting from the combination of different terms. While multi-objective optimisations can indeed be “scalarized”, any such reduction (e.g., constructing a linear combination with multipliers or the quotient of two terms as mentioned by the reviewer) inherently assumes an a priori preference on the sought solutions (e.g., the value of the chosen multipliers). In our design, we explicitly avoid any such decision and instead chose to retain the generality of the full bi-objective problem by proposing an evolution-strategy refinement of the Pareto front, i.e., we evolve the set of all Pareto-efficient solutions. We recognize however that our notation for Eq (2) could have been a source of confusion and have thus put significant effort to improve the clarity of the whole Method in Section 3.1, where we (1) made explicit reference to the Pareto front construction, (2) derived the explicit form of the bi-objective loss for the particular but relevant cases of the search for invariances and adversarial attacks.
>
> In revising this section, we also realized that our mathematical formulation of the optimization problem was confusing for two other reasons. First, to provide a more intuitive explanation of our method, we wrote that the upstream layer k was set to be the input one (i.e., k = 0). Later in the section, we did explain that our method can actually be applied by setting k to any value. However, we understand that framing our method by setting k = 0 was confusing. Thus, in our revised Section 3.1, we have provided from the start a more general formulation where k can take any value. The second issue with our original formulation is that we did not make it clear enough that the optimization, in the downstream target layer l (e.g., the readout layer of Resnet50), was applied to individual units and not to the whole layer. We revised our formulation to clarify this point.
>
> We hope that the reviewer will be able to appreciate these changes to improve the clarity of our mathematical formulation in the **revised  Section 3.1 of our manuscript**.
>
> ### 2) 3-component implementation instead of backpropagating gradients.
>
> The 3-component implementation was necessary to propose a method that can also be applied to the study of biological neural systems. An investigator seeking to understand the neural codes for representing sensory information can record the activity of just a subsample of neurons at a time from one or more brain areas, knowing little else about the system. This "black-box" situation is precisely what our framework is built to handle, similarly to precursor approaches for most exciting image (MEI) synthesis (e.g., Ponce et al., 2019). Hence the need of studying an ANN but with artificial constraints (like no gradients) to make the process more closely resemble a neuroscience application. This scenario requires two key components.
>
> 1) A gradient-free optimizer (CMA-ES) is needed to handle a black-box system, such as a biological brain, where we cannot use gradient-based methods like backpropagation.
>
> 2) A generative model is needed because optimizing an image pixel by pixel requires a search through a vast space that often falls out of the subspace of natural image statistics (which is the relevant regime where visual systems are studied). The generator acts as a crucial prior, constraining the search to a much smaller (e.g., 4096 dimensions), more structured manifold of natural-like images. Operating in this restricted space makes the optimization process more efficient and ensures that the stimuli we discover are perceptually meaningful.
>
> In summary, the 3-component design is necessary for our goal to develop a powerful and flexible framework, which is applicable to virtually any image recognition system (artificial and biological alike), even when gradient-based approaches are not possible. This was already briefly mentioned in the first paragraph of our original Discussion. **To make it more explicit, we have now expanded the Introduction**.

---

> ### Author Response · Authors · 2025-11-20
> **Rebuttal to Empirical issues (1/3)**
>
> ## Empirical results issues.
>
> ### 1) Perturbations can be applied not just to the input level (k=0) but to any layer.
>
> As already mentioned, the upstream layer (k) where the perturbations are applied does not need to be the input one. That is, k does not need to be set to 0, but can take any value. We believe that this misunderstanding arose from our mathematical formulation being not clear enough. Thanks to the reviewer’s comments, **we realized this shortcoming of our formulation and, in our amended manuscript, we have fully revised it (see Section 3.1).**
>
> Here, we take the opportunity of answering the reviewer’s question to expand on the meaning and usefulness of applying perturbations to layers that are not necessarily the input stage. Maximally altering (stretching) the representation of an image in an intermediate upstream layer (i.e., with k > 0) allows a much more general assessment of the invariance field of a readout unit. In fact, the image is transformed along the feature dimensions that are specifically encoded in that layer. When applying this approach across different intermediate layers, one can thus alter the structure of the image along transformation axes that would be impossible to explore by working purely in the pixel space. Ultimately, this is what allowed discovering the hierarchical invariances of Resnet50 readout units in our study (e.g., see Fig. 3 and 4).
>
> ### 2) Reasons to find invariant images for individual units.
>
> We believe that this question stems from a shortcoming of our mathematical formulation in making clear that, in the target layer, our method was applied to find invariant images for individual units and not for the whole layer. As already mentioned, we have corrected our formulation to clarify this point. Here, we take the opportunity to explain the reasons why it makes sense to look for invariant images at the level of individual units.
>
> The notion of finding invariant images is tightly linked to the notion of finding most exciting images (MEIs). MEIs, in turn, are typically computed for individual neurons in artificial neural networks or in the brain (e.g., Ponce et al Cell 2019). Computing a MEI is a way to understand the tuning (or selectivity) of a given unit. The underlying assumption is that a unit has an unknown tuning profile over a feature space. This profile may have one or more peaks (uncovered by computing the MEIs) and these peaks may have a “width” in a high-dimensional space that is more or less broad. Such a width defines the invariance field (or manifold) of the unit, and this is what our method is able to characterize. In other words, working on individual neurons provides a very clear conceptual framework, where both MEIs and invariant images are well defined and meaningful - which is why most feature visualization approaches have been applied to individual neurons in both AI and neuroscience.
>
> For these reasons, in our study, we search for the invariant images of individual readout units and not, collectively, of the whole readout layer. An important implication of this choice is that, although our method implements a dual objective optimization involving two different layers, the optimization acts on all the units of the upstream, stretching layer, while operates on just one unit at the time in the downstream (readout) layer. **Our revised mathematical formulation of the dual-objective optimization in Section 3.1 should have made this clearer.**

---

> ### Author Response · Authors · 2025-11-20
> **Rebuttal to Empirical issues (2/3)**
>
> ### 3) Reasons to use MEIs as references to look for invariant images in Fig. 2
>
> The reason why MEIs were used for this test, instead of natural images, is that, as explained above, the concepts of most exciting images and invariant images are closely related. Assuming that a unit has an underlying, unknown tuning over the image space, finding a MEI allows estimating which visual images the unit is actually tuned to. A MEI can depart considerably from the best natural images, often driving the unit substantially more than the most effective natural image (e.g., Ponce et al., 2019). This is not surprising, given that it is very unlikely that any of the natural images will perfectly match the combination of visual features a unit is tuned for. The same arguments apply to the invariant images. Once a MEI is found, this indicates a location in image space where the peak of a unit’s activation is centered. The next important question is to understand how wide this peak is (i.e., the invariance profile). Our method allows answering this question, by exploring the tuning landscape around the peak and probing the extent to which images can depart from the MEI while maintaining the unit’s activation at a high level.
>
> Nevertheless, our method also works when considering a natural image as a reference. For instance, in the analysis shown in Fig. 4, we have obtained images that are invariant with respect to natural image categories in Imagenet. **Following the suggestion of the reviewer, we have also repeated the tests shown in Fig. 2, but using as references very effective natural images for a set of readout units.** The result is very similar to that obtained with the MEIs. **We have included this analysis in a new supplementary figure (Fig. S4a).**
>
> We also noticed that the reviewer wrote “Figure 2 shows a single example image (an MEI), with a corresponding adversarial and invariant example.” However, we want to clarify that, although the figure only includes visualizations for one example image, the data points shown in Fig. 2 are averages of data obtained from 77 different readout units, each with its own MEI and its own invariant and adversarial images. There was not enough space to visually display all of these images in Fig. 2. To provide the readers with more examples, **we have added a new supplementary figure (Fig. S4b) that shows the MEI, as well as the invariant and adversarial images for 5 example units**. The same figure also shows, for these same units, representative natural images and their associated invariant and adversarial images.
>
> Finally, concerning the question of the reviewer about “why the vertical axis is not Euclidean distance in readout layer”, the reason is that our method was applied to target the activation of individual (readout) units. Therefore, although deviations of the activations of these units from the values measured with the reference MEIs could still be expressed as Euclidean distances, because these deviations are just scalar numbers, we preferred to express them as % decrements relative to activations produced by the MEIs.
>
> ### 4) Failure of our description of the hierarchical invariances in Fig. 3 to fully capture their complexity and variability
> We agree that the terms we used to qualitatively characterize the different kinds of invariances discovered by our method, when the stretching was applied at different depths along the network, fall short at conveying the whole richness and complexity of those invariant images. We used those terms (luminance/contrast changes, texture/color changes, and viewpoint or multiplicity of object changes) only as concise and descriptive labels to highlight at least some of the major qualitative differences that can be appreciated by visually comparing the images. In our revised manuscript, **we have explained more clearly that these are just semantic proxies** for concisely describing image transformations that are, in fact, too rich and complex to be readily captured by a textual description **(see Section 4.2)**.
>
> At the same time, we wish to emphasize that these semantic labels are of limited relevance to the conclusions of our study. What actually matters is the fact that the invariances uncovered by our method at different stretching depths are actually very distinct in the image space. This is quantitatively and objectively demonstrated by the analysis of Fig. 3C, which shows how an SVM classifier is able to faithfully recognize each class of invariant images with high accuracy. This shows how the three sets of invariant images obtained by stretching the representations at three different processing stages along the CNN are highly discriminable. In turn, this demonstrates that our method is able to uncover distinct axes of image variations a readout unit is invariant to, which is one of the major achievements of our approach.

---

> ### Author Response · Authors · 2025-11-20
> **Rebuttal to Empirical issues (3/3)**
>
> ### 5) “Since the combination of the two objective components is not specified, I could not understand the interpretation of the perceptual study. To make comparisons of performance on invariant images generated for different layers, it would seem essential to match the conserved objective (output layer?) ...”
>
> The method for combining the two objectives is a central part of our framework, and we agree that it was not sufficiently clear in the main text. As we describe in detail in Supplementary Material (Section A3), we treat this as a multi-objective optimization problem. At each iteration, candidate solutions are ranked by sorting them into a series of Pareto fronts (PF) in the two-objective space. First, the set of non-dominant solutions is identified as the first front (PF1); this process is then repeated on the remaining candidates to find PF2, and so on. The next optimization step is then guided by these top-ranked solutions (e.g., the top 30% of the candidates) according to the update rules of the CMA-ES algorithm.
>
> Regarding the second point raised by the reviewer (i.e., whether the invariant images found by our algorithm are “invariant enough” for a given readout unit, so as to be meaningful for our perceptual tests with human participants), **we have addressed this issue in Sections A.1 and A.5.1 of the Supplementary Material**. The key question is: what level of activation of the target readout unit should be considered large enough for an image to be considered "invariant"? For example, for a neuron tuned to the class  'goldfish', a reasonable answer is an activation level that is close to or exceeds (like MEIs do) the distribution of responses to other 'goldfish' images, and is very distinct from the distribution of responses to non-'goldfish' images. This is precisely the principle we followed for all our optimizations. **As shown in Fig. S1a**, the target activations we achieve with invariant images greatly exceed those produced by typical natural images of the category that the readout unit encodes (compare the last three sets of boxes to the first set).
>
> This principle also motivates our early-stopping criterion: we consider the optimization to have converged on the invariance manifold when 90% of the candidate images elicit a response that is higher than the response to the single most activating image in the entire ImageNet training set (1.2 million images). This same principle could be easily extended also to neurons that are not tuned to specific categories of stimuli (e.g., biological visual neurons, but also units in hidden layers of a CNN; see Supplementary Material, Section E): invariant images are those that induce in the target neuron activity levels that are equal or exceed the top activations of that neuron within a large set of natural images.  As we noted in our reply to the second concern of the reviewer, this heuristic criterion is particularly valuable because SnS is designed to operate on black-box systems (like biological brains), where theoretical guarantees of convergence are not available, making such data-driven benchmarks essential.
>
> ### 6) Related work
>
> We thank the reviewer for pointing out these highly relevant studies. **We now cite both of them in Section 4.3 of our revised manuscript**, briefly highlighting similarities and differences with our approach. In short: similarly to our use of SnS in the experiments described in Fig. 4, both controversial stimuli and MAD competition are tools that can be used to test model alignment with human perception. However, they differ from SnS regarding which aspects of human-machine alignment are being probed. Specifically:
>
> ·   	SnS generates maximally invariant images for a given target neuron with respect to a specific hierarchically lower representation space. In our experiments, we target mainly readout units, which are the ones that encode specific semantic categories (e.g. the “cat” neuron), and whose invariances thus describe that of the entire network with respect to that image class. In our behavioral experiment, we test if those extreme invariances are interpretable by a human observer. In other words, we test whether these invariances are shared or not between humans and Resnet50.
>
> ·   	Controversial Stimuli and MAD study alignment through model disagreement: they focus on stimuli where the classifications of two different models disagree. The scientific question then becomes: "Faced with this machine-generated controversy, which model's classification aligns better with a human's judgment?".
>
> Based on this comparison, we argue that SnS and Controversial Stimuli/MAD provide distinct and complementary tools that address different aspects of human-machine alignment.

---

### Author Response · Authors · 2025-12-01
**Summary of our rebuttal for the AC**

To ease evaluation by the new area chair, we summarize below the main amendments to the manuscript to address the reviewers’ concerns.

All four reviewers acknowledged the relevance and timeliness of our work as well as the conceptual novelty of our framework. Despite this, reviewer **N9Ge** raised concerns about the method’s clarity and the “anectodal” nature of the experimental results. By contrast, the other 3 reviewers (**oZyT**, **ac3Z**, and **JP16**) underlined the clear presentation of our framework and its experimental application, which was evaluated as “comprehensive” (**oZyT**) and “of high quality and rigor” (**ac3Z**). Moreover, reviewer **JP16** rated our contribution as “excellent” and reviewer **ac3Z** underlined that “the results are not just empirical observations but yield a deep conceptual insight” on network-human and network-network perceptual alignment. Finally, these reviewers praised the “broad significance” (**ac3Z**) of our approach, given that its model-agnostic and gradient-free nature makes it “particularly important with black-box models and neuroscience applications where gradient information is unavailable” (**oZyT**), “which is a major constraint in current gradient-based neuro-visual studies” (**ac3Z**). Supported by these positive assessments, we respectfully disagree with reviewer **N9Ge**, because we believe that the reviewer’s criticisms stem from some fundamental misunderstanding of our method that, we think, we have fully clarified in our rebuttal and revision of the manuscript. More broadly, we have addressed all comments of the reviewers, which we thank for their careful assessment of our study, and we think that the resulting revision has substantially increased the quality and reach of our work.

Overall, the changes to our manuscript  include a total of **5 new supplementary sections, 6 new supplementary figures, and 3 new supplementary tables.** Additionally, 6 sections of the main paper and 4 pre-existing supplementary sections were expanded to include new content derived from the reviewing process. More specifically, the most important changes we made to address each reviewer’s concerns are listed below.

---

> ### Author Response · Authors · 2025-12-01
> **Summary of our main revisions for the AC**
>
> ### Reviewer N9Ge (amendments to the manuscript in red font)
> - **Methods clarification**: **we substantially revised the Methods Section (3.1)** to more clearly explain two aspects that **had not been evident to the Reviewer, leading to their main criticisms of our work**: 1) the SnS bi-objective optimization, that evolves the Pareto front rather than using a linear combination of losses; and 2) the fact that  invariant/adversarial perturbations can be applied not only in pixel space but at any hidden layer.
>
> - **Clarified motivation for our 3-component design** (evolutionary optimizer + subject network + generator). This is required to make SnS applicable not only to ANNs but also to biological systems, where gradients are unavailable (see revised Section 1).
>
> - We clarified why Most Exciting Images (MEIs) were used as references in SnS image synthesis. Moreover, **we added new findings indicating that SnS can be used equally well with natural images as references  (new Fig. S4 in the new Supplementary Section C)**, where we also provided more extensive SnS image synthesis examples.
>
> ### Reviewer oZyT (amendments to the manuscript in brown font)
> - **We discussed the dependence of SnS running time** on network size and code dimensionality, noting that it remains practical and comparable to gradient-based metamer synthesis (Feather et al., 2023) on our machine (additions to Supplementary Sections A.1 and A.6).
> - **We analyzed the geometry of SnS invariance manifolds** for a L2-robust ResNet-50 readout neuron, using both a standard linear method (PCA) and a state-of-the-art nonlinear approach (2NN-ID) to estimate the intrinsic dimension of data manifolds (see the **new Supplementary Section H and new Fig. S9**).
> - **We carried out a new analysis on SnS convergence**, adding convergence curves **in the new Fig. S1b**, discussing the important question of **what is needed for a stimulus to be considered invariant**. We also showed that optimizations that do not converge still yield invariant images (**see the new Table S1 in Section A.5.1**).
> - **We compared SnS invariances to those yielded by a state-of-the-art gradient-based method: model metamers** (Feather et al., 2023). We showed fundamental differences in their alignment properties (**new Tables S3 and S4 in the new Supplementary Section I**). We also noted that metamer-like stimuli emerge as a special case of the SnS objective (see the “squeeze-down’’ variant in the **new Fig. S5**), indicating that SnS can generalize metamer synthesis in a gradient-free framework.
>
> ### Reviewer ac3Z (amendments to the manuscript in blue font)
> - **We compared the suggested literature (EITs and MIRCs) to SnS (Section 4.3).**
> - We discussed the possibility of using different generator models, such as large-scale GAN and diffusion models, highlighting the pros and cons of these approaches (see additions to **Section 5**).
> - **We replicated the experiment in Fig. 4c with a L_inf generator ResNet50**. Results are described in the **new dedicated Supplementary Section J (see the new Fig. S10)** and commented in the main text (**Section 4.3**).
> - **We have added a dedicated visualization of the final Pareto fronts** for several representative units in the **new supplementary Fig. S0.**
> - We argued that **SnS has been validated across diverse architectures (CNNs and ViTs), parameter scales, and training objectives (L2 robust and standard training)**, and, being model-agnostic, is expected to generalize to even larger networks (see additions to **Section 5**).
> - We explained that validating SnS against fMRI or EEG would require dedicated encoding models, which is beyond the scope of the current work. However, we argue that our current characterization of SnS holds multiple important insights for application to neuroscience. First, we characterized invariances on networks whose activations reliably predict neural activity in the primate visual system, as shown by metrics such as Brainscore (Schrimpf et al., 2018). Second, we showed that SnS can be effectively applied to intermediate convolutional units (Supplementary Section E) and, third, that SnS hierarchical invariances remain robust even when the upstream representations stretched are heavily subsampled, as in neurophysiology experiments (Supplementary Section F).
>
> ### Reviewer JP16 (amendments to the manuscript in violet font)
> - We clarified the basis for our claim that “L2 adversarial training fails to increase the interpretability of high-level invariances” (see additions to **Section 4.3**)
> - We repeated the experiment in Fig. 2 with the **single loss variants** suggested by the reviewer. We believe that this has significantly strengthened our approach as **none of the four single-loss strategies optimized the dual trade-off typical of SnS** (see the **new supplementary section D and Fig. S5** and the addition to **Section 4.1**).

---

### Meta-Review · Area_Chair_G62E · 2025-12-19

**Summary:**

This paper introduces Stretch-and-Squeeze (SnS), a gradient-free, model-agnostic framework for characterizing visual invariances and adversarial vulnerabilities in neural networks. The method uses bi-objective optimization to find image perturbations that maximally alter representations at one layer (stretch) while preserving activations at another layer (squeeze). Applied to CNNs and Vision Transformers, the authors discover hierarchical invariances and find that adversarially-trained networks show decreasing interpretability of high-level invariances to humans, despite good pixel-level alignment.

**Reviewer Concerns**
The paper received mixed reviews with scores of 2, 6, 6, and 6 from reviewers N9Ge, oZyT, ac3Z, and JP16 respectively.

**Reviewer Concerns:**

**Addressed concerns:**

* Method clarity (N9Ge): The authors substantially revised Section 3.1 to clarify the bi-objective Pareto front optimization and the hierarchical nature of layer selection. The mathematical formulation is now explicit about operating on individual units in the target layer while perturbing entire representations upstream.
* Computational cost (oZyT): Analysis added showing ~120 seconds per optimization run, with discussion of scalability and parallelization opportunities.
* Comparison to gradient-based methods (oZyT, ac3Z): Comprehensive comparison to model metamers (Feather et al. 2023) added.
* Invariance manifold geometry (oZyT): New analysis using PCA and 2NN-ID showing nonlinear manifold structure with dimensionality patterns consistent with known properties of CNN representations.
* Convergence analysis (ac3Z): Convergence curves added (Fig. S1b), with detailed analysis of non-converging runs showing they still reach functionally relevant activation regimes.
* Alternative robustness regimes (ac3Z): $L_\infty$ robust network experiments completed (Fig. S10), revealing different invariance patterns than L2 robustness.
* Single-loss ablations (JP16): Four single-loss variants tested, confirming the necessity of the dual-loss formulation.

**Remaining concerns**
* N9Ge's characterization as "anecdotal": I disagree with this assessment. The experiments involve 77 readout units with 10 seeds each (770 runs per condition), human behavioral studies with 25 participants and statistical analysis, and validation across 8 different architectures. This is comprehensive, not anecdotal.
* Lack of biological validation: While submitted to the neuroscience applications track, the paper lacks in vivo neural data. While the authors acknowledge that, they validated on brain-like models (Brain-Score benchmarked architectures), demonstrate applicability to intermediate layers with localized receptive fields, and show robustness to representation subsampling—all directly relevant for future neuroscience applications.

**Reviewer Scores:**

Three reviewers converged on marginal acceptance (score 6), while one recommended rejection (score 2). However, N9Ge's  (score 2) concerns possibly originate from misunderstandings of the method that were thoroughly addressed in the rebuttal. The other reviewers explicitly acknowledged the novelty, significance, and practical value of the work.

---

### Decision · Program_Chairs · 2026-01-26

Accept (Poster)